# A Hierarchical Training Paradigm for Antibody Structure-sequence Co-design

**Fang Wu**[*]
Tsinghua University
Beijing, China

**Stan Z. Li**
Westlake University
Hangzhou, China

## Abstract

Therapeutic antibodies are an essential and rapidly expanding drug modality. The binding specificity between antibodies and antigens is decided by complementarity-determining regions (CDRs) at the tips of these Y-shaped proteins. In this paper, we propose a **h**ierarchical **t**raining **p**aradigm (HTP) for the antibody sequence-structure co-design. HTP consists of four levels of training stages, each corresponding to a specific protein modality within a particular protein domain. Through carefully crafted tasks in different stages, HTP seamlessly and effectively integrates geometric graph neural networks (GNNs) with large-scale protein language models to excavate evolutionary information from not only geometric structures but also vast antibody and non-antibody sequence databases, which determines ligand binding pose and strength. Empirical experiments show that HTP sets the new state-of-the-art performance in the co-design problem as well as the fix-backbone design. Our research offers a hopeful path to unleash the potential of deep generative architectures and seeks to illuminate the way forward for the antibody sequence and structure co-design challenge.

## 1   Introduction

Antibodies, known as immunoglobulins (Ig), are large Y-shaped proteins that the immune system uses to identify and neutralize foreign objects such as pathogenic bacteria and viruses [1, 2]. They recognize a unique molecule of the pathogen, called an antigen. As illustrated in Figure 1 (a), each tip of the "Y" of an antibody contains a paratope that is specific for one particular epitope on an antigen, allowing these two structures to bind together with precision. Notably, the binding specificity of antibodies is largely determined by their complementarity-determining regions (CDRs). Consequently, unremitting efforts have been made to automate the creation of CDR subsequences with desired constraints of binding affinity, stability, and synthesizability [3]. However, the search space is vast, with up to $20^L$ possible combinations for a $L$-length CDR sequence. This makes it infeasible to solve the protein structures and then examine their corresponding binding properties via experimental approaches. As a remedy, a group of computational antibody design mechanisms has been introduced to accelerate this filtering process.

Some prior studies [4, 5] prefer to generate only 1D sequences, which has been considered suboptimal because it lacks valuable geometric information. Meanwhile, since the target structure for antibodies is rarely given as a prerequisite [6], more attention has been paid to co-designing the sequence and structure. One conventional line of research [7–9] resorts to sampling protein sequences and structures on the complex energy landscape constructed by physical and chemical principles. But it is found to be time-exhausted and vulnerable to being trapped in local energy optima. Another line [10–12] relies on deep generative models to simultaneously design antibodies' sequences and structures. They take advantage of the most advanced geometric deep learning (DL) techniques and can seize

---

[*]Corresponding Authors, emails: `fw2359@columbia.edu`.

37th Conference on Neural Information Processing Systems (NeurIPS 2023).

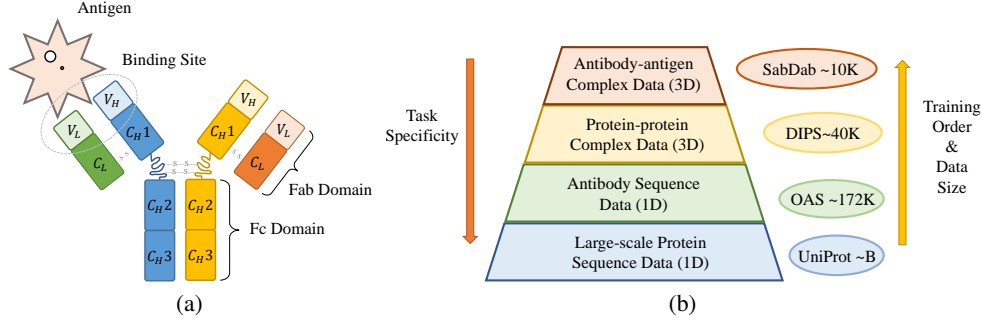

Figure 1: (a) Schematic structure of an antibody bonded with an antigen (figure modified from Wikipedia). (b) The workflow overview of our hierarchical training paradigm (HTP).

higher-order interactions among residues directly from the data [13]. Their divergence mainly lies in their generative manner. For instance, early works [10, 11] adopt an iterative fashion, while successors [14, 15] employ full-shot generation. Most utilize the traditional translation framework, while some [12] leverage the diffusion denoise probabilistic model.

Despite this fruitful progress, the efficacy of existing co-design methods is predominantly limited by the small number of antibody structures. The Structural Antibody Database (SAbDab)[16] and RAbD[9] are two widely used datasets in the field. After eliminating structures without antigens and removing duplicates, SAbDab comprises only a few thousand complex structures, whereas RAbD consists of 60 complex structures. These numbers are orders of magnitude lower than the data sizes that can inspire major breakthroughs in DL areas [17, 18]. Consequently, deep-generative models fail to benefit from large amounts of 3D antibody-antigen complex structures and are of limited sizes, or otherwise, overfitting may occur.

To address this issue, in this paper, we propose a hierarchical training paradigm (HTP), a novel unified prototype to exploit multiple biological data resources, and aim at fully releasing the potential of geometric graph neural networks (GGNNs) [19, 20] for the sequence and structure co-design problem. Explicitly, HTP consists of four distinct training ranks: single-protein sequence level, antibody sequence level, protein-protein complex structure level, and antibody-antigen complex structure level. These steps are itemized as the data abundance declines, but the task specificity increases, as depicted in Figure 1 (b). Alongside them, we present various pretraining objectives for the first three levels to mine correlated evolutionary patterns, which benefit the co-design task in the final stage. Specifically, we first pretrain the protein language models (PLMs) on tremendous single-protein sequences to obtain general representations and fine-tune them on antibody sequences to capture more condensed semantics. For 3D geometry, we invented a pocket-painting task to exploit the protein-protein complex structures and simulate the CDR generation process. After that, we combine the marvelously pretrained PLMs and the well-pretrained GGNNs to co-design the expected antibodies. Our study provides a promising road to excavate the power of existing deep generative architectures and hopes to shed light on the future development of antibody sequence and structure co-design. The contributions of our work can be summarized as follows.

- First, we equip GGNNs with large-scale PLMs to bridge the gap between protein databases of different modalities (*i.e.*, 1D sequence, and 3D structure) for the antibody co-design problem.

- Second, we design four distinct levels of training tasks to hierarchically incorporate protein data of different domains (*i.e.*, antibodies, and non-antibodies) for the antibody co-design challenge. HTP breaks the traditional co-design routine that separates proteins of different domains and extends antibody-antigen complex structures to broader databases.

- Comprehensive experiments have been conducted to indicate that each stage of HTP significantly contributes to the improvements in the capacity of the DL model to predict more accurate antibody sequences and restore its corresponding 3D structures. To be explicit, HTP brings a rise of 78.56% in the amino acid recovery rate (AAR) and a decline of 41.97% in structure prediction error for the sequence-structure co-design. It also leads to an average increase of 26.92% in AAR for the fixed-backbone design problem.

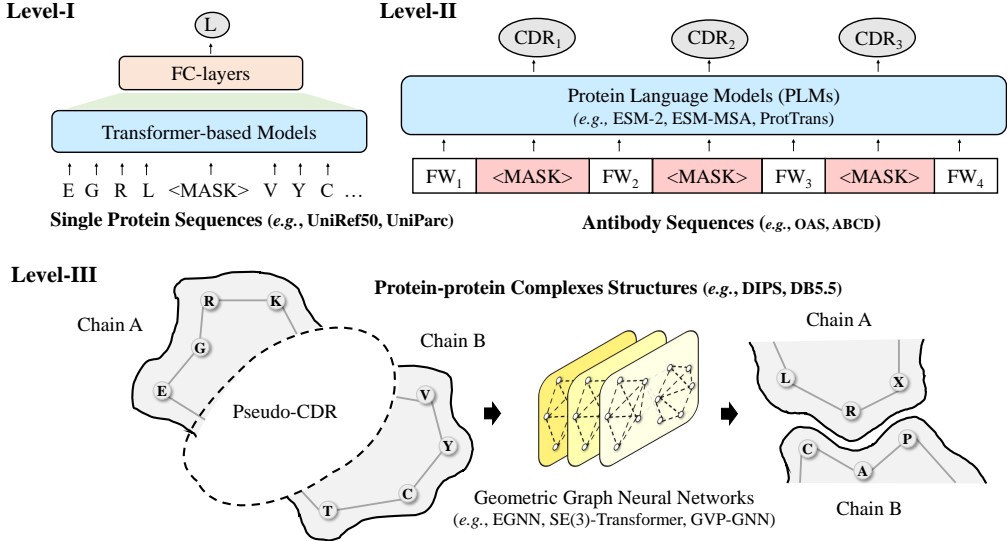

Figure 2: The illustration of the first three stages of our hierarchical training mechanism for antibody sequence-structure co-design. In **level I**, a Transformer-based language model is trained by masked language modeling on a large number of single protein sequences to extract general-purposed representations. In **level II**, the Transformer-based PLM is then further fine-tuned on specific antibody sequences from databases like OAS or ABCD. CDRs are all masked and require recovery, with other framework regions reserved. In **level III**, GGNNs are asked to predict both the sequence and structure of the pseudo-CDR on protein-protein complex structure databases. The residue features are based on PLMs gained in the previous step.

## 2 Methods

This section is organized as follows. Subsection 2.1 describes the background knowledge and mathematical formulation of the co-design problem. Subsection 2.4 introduces the backbone architecture to encode the geometric structure of protein-protein complexes as well as antibody-antigen complexes in the 3D space. Subsection 2.2 to 2.6 concentrate on explaining the four different levels of our HTP.

### 2.1 Preliminary

**Background** An antibody is a Y-shaped protein with two symmetric sets of chains, each consisting of a heavy chain and a light chain. Each chain has one variable domain (VH/VL) and some constant domains. The variable domain can be further divided into a framework region (FW) and three CDRs. Notably, CDRs in heavy chains contribute the most to the antigen-binding affinity and are the most challenging to characterize.

**Notations and Task Formulation** We represent each antibody-antigen complex as a heterogeneous graph $\mathcal{G}_{LR}$. It is made up of two spatially aggregated components, *i.e.*, the antibody and antigen denoted as $\mathcal{G}_L = \{\mathcal{V}_L, \mathcal{E}_L\}$ and $\mathcal{G}_R = \{\mathcal{V}_R, \mathcal{E}_R\}$, respectively. $\mathcal{G}_L$ and $\mathcal{G}_R$ use residues as nodes with numbers of $N_L$ and $N_R$ separately. The node locations $\mathbf{x}_L \in \mathbb{R}^{N_L \times 3}$ and $\mathbf{x}_R \in \mathbb{R}^{N_R \times 3}$ are defined as their corresponding $\alpha$-carbon coordinate, and are associated with the initial $\psi_h$-dimension roto-translation invariant features $\mathbf{h}_L \in \mathbb{R}^{N_L \times \psi_h}$ and $\mathbf{h}_R \in \mathbb{R}^{N_R \times \psi_h}$ (*e.g.* residue types, electronegativity). CDRs are subgraphs of $\mathcal{G}_L$ and can be divided into $\mathcal{G}_{HC} = \{\mathcal{V}_{HC}, \mathcal{E}_{HC}\}$ and $\mathcal{G}_{LC} = \{\mathcal{V}_{LC}, \mathcal{E}_{LC}\}$, which belong to the heavy chain and the light chain, respectively. We assume that $\mathcal{G}_{HC}$ and $\mathcal{G}_{LC}$ have $N_{HC}$ and $N_{LC}$ residues. Besides, it is worth noting that the distance scales between the internal and external interactions are very different. Based on this fact, we strictly distinguish the interaction within and across two graphs $\mathcal{G}_R$ and $\mathcal{G}_L$ as and $\mathcal{E}_L \cup \mathcal{E}_R$ and $\mathcal{E}_{LR}$, individually. This implementation avoids the underutilization of cross-graph edges' information due to implicit positional relationships between the antibody and the antigen [21].

In this work, we hypothesize that the antigen structure and the antibody framework are known, aiming to design CDR with more efficacy. Formally, our goal is to jointly model the distribution of CDR in the heavy chain given the structure of the remaining antibody-antigen complex as $p(\mathcal{G}_{HC}|\mathcal{G}_{LR}-\mathcal{G}_{HC})$ or in the light chain as $p(\mathcal{G}_{LC}|\mathcal{G}_{LR} - \mathcal{G}_{LC})$. Since the heavy chain plays a more critical role in determining antigen binding affinity, we take $\mathcal{G}_{HC}$ as the target example in the following content, but design both heavy and light chains in the experiment section 3.1.

## 2.2 Single-protein Sequence Level

The idea that biological function and structures are recorded in the statistics of protein sequences selected through evolution has a long history [22]. Unobserved variables that determine the fitness of a protein, such as structure, function, and stability, leave a mark on the distribution of the natural sequence observed [23]. To uncover that information, a group of PLMs has been developed at the scale of evolution, including the series of ESM [22] and ProtTrans [24]. They are capable of capturing information about secondary and tertiary structures and can be generalized across a broad range of downstream applications. Recent studies [25] also demonstrate that equipping GGNNs with pretrained language models can end up with a stronger capacity. Accordingly, we adopt an ESM-2 with 150M parameters to extract per-residue representations, denoted as $\mathbf{h}'_L \in \mathbb{R}^{N_L \times \psi_{PLM}}$ and $\mathbf{h}'_R \in \mathbb{R}^{N_R \times \psi_{PLM}}$, and use them as input node features. Here $\psi_{PLM} = 640$ and we use $\Phi$ to denote the trainable parameter set of the language model.

Noteworthily, ESM-2 supplies plenty of options with different model sizes, ranging from 8M to 15B, and a persistent improvement has been found as the model scale increases. However, the trajectory of improvement becomes relatively smooth after the scale reaches $10^8$. Therefore, for the sake of computational efficiency, we select the 150M version of ESM-2, which performs comparably with the 650M parameters of the ESM-1b model [22]. As declared by Wu et al. [26], incompatibility exists between the experimental structure and its original amino acid sequence (*i.e.*, FASTA sequence). For simplicity, we obey Wu et al. [26]'s mode, which uses the fragmentary sequence directly as the substitute for the integral amino acid sequence and forwards it to the PLMs.

## 2.3 Antibody Sequence Level

Though PLMs have achieved great progress within protein informatics, their protein representations are generally purposed. Noticeably, the residue distribution of antibodies is significantly different from non-antibodies. Over the past decades, billions of antibodies have been sequenced [27], which enables the training of a language model specifically for the antibody domain [28]. Several researchers have recognized this problem and present models such as AntiBERTa [29] and AbLang [30] to decipher the biology of disease and the discovery of novel therapeutic antibodies. Nonetheless, those pretrained antibody language models have some intrinsic flaws and may not be suitable to apply in our sequence-structure co-design problem immediately.

First and foremost, they are directly pretrained on antibody sequence datasets and fail to exploit the vast amounts of all protein sequences. Second, the existing pretrained antibody language models are all on a small scale. For example, AntiBERTa consists of 12 layers with 86M parameters. Last but not least, both AntiBERTa and AbLang regard the heavy and light chains individually, and AbLang even trains two different models for them. This approach is precluded from encoding a comprehensive and integral representation of the whole antibody. More importantly, they are pretrained on antibodies and cannot be perfectly generalized to extract representations of antigens, which is necessary for analyzing the interactions between antibodies and antigens.

To avoid their drawbacks, we chose to fine-tune ESM on available antibody sequence datasets and considered both antibody and antigen sequences. Following AbLang [30], we leverage the Observed Antibody Space database (OAS) [31] and subsequent update [32]. OAS is a project that collects and annotates immune repertoires for use in large-scale analysis. It contains over one billion sequences, from over 80 different studies. These repertoires cover diverse immune states, organisms, and individuals. Additionally, Olsen et al. [30] has observed in OAS that approximately 80% sequences lack more than one residue at the N-terminus, nearly 43% of them are missing the first 15 positions, and about 1% contain at least one ambiguous residue for each sequence. Remarkably, OAS contains both unpaired and paired antibody sequences and we utilize only paired ones. To better align with our

co-design target, we implement masked language modeling (MLM) and mask residues in all CDRs (*i.e.*, VH, and VL) simultaneously to increase the task difficulty.

## 2.4 Geometric Graph Neural Networks

To capture 3D interactions of residues in different chains, we adopt a variant of equivariant graph neural network (EGNN) [33] to act on this heterogeneous 3D antibody-antigen graph. The architecture has several key improvements. First, it consists of both the *intra-* and *inter-* message-passing schemes to distinguish interactions within the same graph and interactions between different counterparts. Second, it only updates the coordinates of residues in CDR, *i.e.*, the part that is to be designed, while the positions of other parts are maintained as unchangeable. Last, it uses features from the pretrained PLMs as the initial node state rather than a randomized one. Here, all modules are E(3)-equivariant.

The *l*-th layer of our backbone is formally defined as the following:

$$\mathbf{m}_{j \to i} = \phi_e \left( \mathbf{h}_i^{(l)}, \mathbf{h}_j^{(l)}, d\left( \mathbf{x}_i^{(l)}, \mathbf{x}_j^{(l)} \right) \right), \forall e_{ij} \in \mathcal{E}_L \cup \mathcal{E}_R, \tag{1}$$

$$\boldsymbol{\mu}_{j \to i} = a_{j \to i} \mathbf{h}_j^{(l)} \cdot \phi_d \left( d\left( \mathbf{x}_i^{(l)}, \mathbf{x}_j^{(l)} \right) \right), \forall e_{ij} \in \mathcal{E}_{LR}, \tag{2}$$

$$\mathbf{h}_i^{(l+1)} = \phi_h \left( \mathbf{h}_i^{(l)}, \sum_j \mathbf{m}_{j \to i}, \sum_{j'} \boldsymbol{\mu}_{j' \to i} \right), \tag{3}$$

where $d(.,.)$ is the Euclidean distance function. $\phi_e$ is the edge operation and $\phi_h$ denotes the node operation that aggregates the *intra*-graph messages $\mathbf{m}_i = \sum_j \mathbf{m}_{j \to i}$ and the cross-graph message $\boldsymbol{\mu}_i = \sum_{j'} \boldsymbol{\mu}_{j' \to i}$ as well as the node embeddings $\mathbf{h}_i^{(l)}$ to acquire the updated node embedding $\mathbf{h}_i^{(l+1)}$. $\phi_d$ operates on the inter-atomic distances. $\phi_e$, $\phi_h$ and $\phi_d$ are all multi-layer perceptrons (MLPs). Besides that, $a_{j \to i}$ is an attention weight with trainable MLPs $\phi^q$ and $\phi^k$, and takes the following form as:

$$a_{j \to i} = \frac{\exp\left( \left\langle \phi^q\left( \mathbf{h}_i^{(l)} \right), \phi^k\left( \mathbf{h}_j^{(l)} \right) \right\rangle \right)}{\sum_{j'} \exp\left( \left\langle \phi^q\left( \mathbf{h}_i^{(l)} \right), \phi^k\left( \mathbf{h}_{j'}^{(l)} \right) \right\rangle \right)}. \tag{4}$$

As for coordinate iterations, we note that residues located in CDRs (*i.e.*, $\mathcal{G}_{HC}$) are the sole constituent that needs spatial transformation. On the contrary, the position of the remaining piece (*i.e.*, $\mathcal{G}_{LR} - \mathcal{G}_{HC}$) is ascertainable. If we change the coordinate of $\mathcal{G}_{LR} - \mathcal{G}_{HC}$, its conformation can be disorganized and irrational from physical or biochemical perspectives. Therefore, it is reasonable to stabilize $\mathcal{G}_{LR} - \mathcal{G}_{HC}$ in each layer and simply alter $\mathcal{G}_{HC}$. Mathematically,

$$\mathbf{x}_i^{(l+1)} = \begin{cases} \mathbf{x}_i^{(l)} + \frac{1}{|\mathcal{N}_i|} \sum_{i \in \mathcal{N}_i} \left( \mathbf{x}_j^{(l)} - \mathbf{x}_j^{(l)} \right) \phi_x(i,j), & \text{if } \mathbf{x}_i \in \mathcal{G}_{HC} \\ \mathbf{x}_i^{(l)}, & \text{otherwise} \end{cases}, \tag{5}$$

where $\mathcal{N}_i$ denotes the neighbors of node $i$ and we take the mean aggregation to update the coordinate for each movable node. $\phi_x$ varies according to whether the edge $e_{ij}$ represent *intra*-graph connectivity or cross-graph connectivity. In particular, $\phi_x = \phi_m(\mathbf{m}_{i \to j})$ if $e_{ij} \in \mathcal{E}_L^{(t)} \cup \mathcal{E}_R^{(t)}$. Otherwise, $\phi_x = \phi_\mu(\boldsymbol{\mu}_{i \to j})$ when $e_{ij} \in \mathcal{E}_{LR}^{(t)}$, where $\phi_m$ and $\phi_\mu$ are two different functions to deal with different types of messages. Then, $\phi_x$ is left multiplied with $\mathbf{x}_i^{(l)} - \mathbf{x}_j^{(l)}$ to keep the direction information. Equation 5 takes as input the edge embedding $\mathbf{m}_{i \to j}$ or $\boldsymbol{\mu}_{i \to j}$ as a weight to sum all relative coordinate $\mathbf{x}_i^{(l)} - \mathbf{x}_j^{(l)}$ and output the renewed coordinates $\mathbf{x}_i^{(l+1)}$.

To summarize, the *l*-th layer of our architecture ($l \in [L]$) takes as input the set of atom embeddings $\left\{ \mathbf{h}_L^{(l)}, \mathbf{h}_R^{(l)} \right\}$, and 3D coordinates $\left\{ \mathbf{x}_L^{(l)}, \mathbf{x}_R^{(l)} \right\}$. Then it outputs a transformation on $\left\{ \mathbf{h}_L^{(l)}, \mathbf{h}_R^{(l)} \right\}$ as well as coordinates of residues on the CDRs, that is, $\mathbf{x}_{HC}^{(l)}$. Concisely, $\mathbf{h}_L^{(l+1)}, \mathbf{x}_{HC}^{(l+1)}, \mathbf{h}_R^{(l+1)} = \text{GGNN}^{(l)} \left( \mathbf{h}_L^{(l)}, \mathbf{x}_L^{(l)}, \mathbf{h}_R^{(l)}, \mathbf{x}_R^{(l)} \right)$, while the coordinates of other non-CDR parts remain the same as in the last layer as $\mathbf{x}_L^{(l+1)} \cup \mathbf{x}_R^{(l+1)} \backslash \mathbf{x}_{HC}^{(l+1)} = \mathbf{x}_L^{(l)} \cup \mathbf{x}_R^{(l)} \backslash \mathbf{x}_{HC}^{(l)}$. We assign $\Theta$ as the trainable parameter set of the whole GGNN architecture.

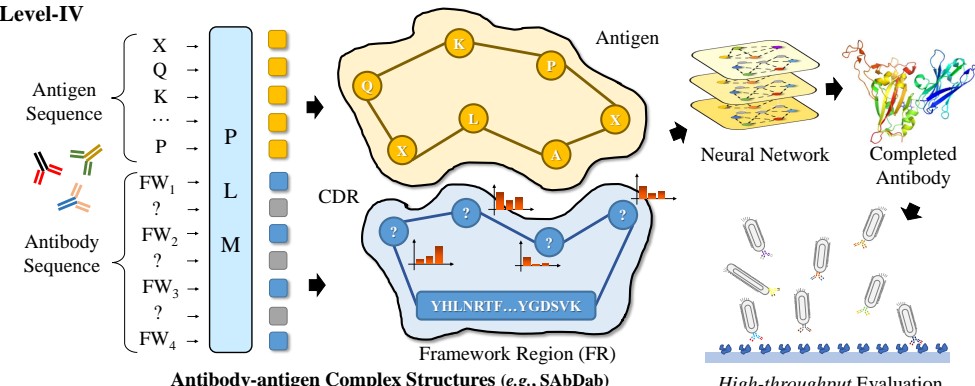

Figure 3: The final stage of our hierarchical training mechanism for antibody sequence-structure co-design. In **level IV**, both the pretrained language model and geometric graph neural networks are employed to implement the co-design task with the parameters of the pretrained language model fixed. At last, the *in silico* antibody is experimentally validated via high-throughput binding quantification.

## 2.5  Protein-protein Complex Structure Level

Apart from empowering PLMs with tremendous protein sequences, plenty of protein-protein complex structures stay uncultivated, which can be harnessed to promote the geometric backbone architecture. Despite the emergence of several structure-based pretraining methods in the biological domain [21, 34], it is not trivial to apply them to our antibody design problem because all previous studies focus on single protein structures. In order to enable GGNNs to encode the general docking pattern between multiple proteins, we propose the pocket inpainting task.

Specifically, we used the Database of Interacting Protein Structures (DIPS) [35]. DIPS is a much larger protein complex structure dataset than existing antibody-antigen complex structure datasets and is derived from the Protein Data Bank (PDB) [36]. In DIPS, each complex $\mathcal{G}_{LR}$ has two sub-units $\mathcal{G}_L$ and $\mathcal{G}_R$. We calculate the distance between each amino acid of these two substructures and select residues whose minimum distance to the counterpart substructure is less than the threshold $\epsilon_P = 8\text{Å}$ as the pocket $\mathcal{G}_P$. Mathematically, the pocket nodes $\mathcal{V}_P$ is written as follows:

$$\mathcal{V}_P = \left\{ v_{L,i} \mid \min_{j=1}^{N_L} d(\mathbf{x}_{Li}, \mathbf{x}_{R,j}) < \epsilon_P \right\} \cup \left\{ v_{R,i} \mid \min_{j=1}^{N_R} d(\mathbf{x}_{R,i}, \mathbf{x}_{L,j}) < \epsilon_P \right\}, \tag{6}$$

then our target is to retrieve $\mathcal{G}_P$ given the leftover $\mathcal{G}_{LR} - \mathcal{G}_P$. Here, We follow the official split based on PDB sequence clustering at a 30% sequence identity level to ensure little contamination between sets. It results in train/val/test of 87,303/31,050/15,268 complex samples.

## 2.6  Antibody-antigen Complex Structures Level

After the preceding levels of preparation, it is time to co-design the antibody. Given an antigen $\mathcal{G}_R$ and a fractional antibody $\mathcal{G}_L - \mathcal{G}_{HC}$, we first employ the well-trained PLM $\Phi$ to attain per-residue node features $\mathbf{h}_R^{(0)}$ and $\mathbf{h}_L^{(0)}$, where the nodes in the unknown part $\mathcal{G}_{HC}$ are tagged as a masked token. Then both features and coordinates are fed into the well-trained geometric encoder to unravel the CDR sequences and structures in a concurrent way, *i.e.*, $\mathbf{h}_L^{(L)}, \mathbf{x}_{HC}^{(L)}, \mathbf{h}_R^{(L)} = \text{EGNN}_\Theta \left( \mathbf{h}_L^{(0)}, \mathbf{x}_L^{(0)}, \mathbf{h}_R^{(0)}, \mathbf{x}_R^{(0)} \right)$. Eventually, we use an MLP $\phi_o$ and a Softmax operator as the classifier to output the probability distribution of residue types as $p_i = Softmax \left( \phi_o \left( \mathbf{h}_i^{(L)} \right) \right) \in \mathbb{R}^{20}$ for $v_i \in \mathcal{V}_{HC}$ .

**CDR Coordinates Initialization.**    How to initialize the positions of residues in CDRs is of great importance to the co-design problem, and there is no consensus among different approaches. HERN [11] tries two kinds of strategies. One strategy is to randomly initialize all coordinates by adding a small Gaussian noise around the center of the epitope as $\mathbf{x}_i^{(0)} = \frac{1}{N_R} \sum_{j \in \mathcal{G}_R} \mathbf{x}_j + \epsilon, \epsilon \sim \mathcal{N}(0, 1)$. The other is to predict the pairwise distance instantly $\mathbf{D}^{(N_L + N_R) \times (N_L + N_R)}$ between paratope and epitope atoms

Table 1: Results of sequence and structure co-deign on SAbDab.

| Model | CDR-H1 | | | CDR-H2 | | | CDR-H3 | | |
|---|---|---|---|---|---|---|---|---|---|
| | AAR (%) ↑ | RMSD ↓ | TM-Score ↑ | AAR (%) ↑ | RMSD ↓ | TM-Score ↑ | AAR (%) ↑ | RMSD ↓ | TM-Score ↑ |
| RAbD | 20.63±1.6 | 3.56±0.05 | 0.9206±0.007 | 27.80±0.8 | 2.85±0.09 | 0.9253±0.010 | 21.73±0.7 | 4.58±0.13 | 0.8916±0.012 |
| C-RGNN | 40.39±3.2 | 1.98±0.02 | 0.9380±0.003 | 33.36±1.7 | 1.32±0.05 | 0.9507±0.005 | 21.89±1.5 | 3.59±0.16 | 0.9187±0.011 |
| MEAN | 43.80±2.5 | 1.84±0.04 | 0.9411±0.008 | 37.18±1.5 | 1.27±0.04 | 0.9522±0.007 | 22.56±1.7 | 3.44±0.18 | 0.9248±0.009 |
| HERN | 48.42±2.7 | 1.69±0.04 | 0.9472±0.005 | 41.53±2.1 | 1.26±0.03 | 0.9531±0.006 | 25.73±1.4 | 3.02±0.11 | 0.9340±0.004 |
| DiffAb | 52.82±0.9 | 1.51±0.01 | 0.9658±0.001 | 45.95±2.3 | 1.24±0.01 | 0.9588±0.002 | 27.04±2.8 | 2.89±0.15 | 0.9417±0.008 |
| HTP | 81.33±1.6 | 0.49±0.02 | 0.9829±0.006 | 67.77±1.3 | 0.53±0.02 | 0.9860±0.004 | 40.98±1.5 | 2.06±0.03 | 0.9621±0.005 |

| Model | CDR-L1 | | | CDR-L2 | | | CDR-L3 | | |
|---|---|---|---|---|---|---|---|---|---|
| | AAR (%) ↑ | RMSD ↓ | TM-Score ↑ | AAR (%) ↑ | RMSD ↓ | TM-Score ↑ | AAR (%) ↑ | RMSD ↓ | TM-Score ↑ |
| RAbD | 35.11±1.0 | 1.88±0.01 | 0.9458±0.002 | 27.82±0.6 | 1.35±0.02 | 0.9611±0.009 | 23.73±0.5 | 2.14±0.06 | 0.9247±0.010 |
| C-RGNN | 41.44±2.5 | 2.06±0.02 | 0.9326±0.008 | 36.71±4.3 | 1.26±0.01 | 0.9652±0.006 | 33.80±4.8 | 1.95±0.06 | 0.9308±0.008 |
| MEAN | 47.69±2.3 | 1.87±0.02 | 0.9461±0.006 | 39.42±3.5 | 1.24±0.01 | 0.9647±0.008 | 35.18±2.6 | 1.84±0.05 | 0.9370±0.005 |
| HERN | 55.24±2.7 | 1.63±0.02 | 0.9502±0.003 | 46.02±4.1 | 1.18±0.02 | 0.9712±0.004 | 37.28±4.1 | 1.77±0.07 | 0.9389±0.002 |
| DiffAb | 62.71±1.2 | 1.48±0.01 | 0.9637±0.002 | 52.10±3.6 | 1.11±0.06 | 0.9780±0.014 | 43.62±2.6 | 1.65±0.05 | 0.9447±0.004 |
| HTP | 91.13±1.0 | 0.67±0.04 | 0.9869±0.005 | 89.80±0.8 | 1.03±0.05 | 0.9846±0.008 | 73.82±1.1 | 0.78±0.04 | 0.9704±0.003 |

and reconstruct atom coordinates from this distance matrix. The results show that the former performs better. DiffAB [12] initializes them from the standard normal distribution as $\mathbf{x}_i^{(0)} \sim \mathcal{N}(\mathbf{0}, \mathbf{I}_3)$. MEAN [14] leverage the even distribution between the residue right before CDRs and the one right after CDRs as the initial positions.

All of the above initialization mechanisms have corresponding drawbacks. To be specific, HERN insufficiently considers the context information since it only characterizes the shape of the epitope and ignores the incomplete antibody structure $\mathcal{G}_{LR} - \mathcal{G}_{HC}$. The initialization of normal distribution is only suitable in diffusion-based models. Moreover, our empirical experiments observe the least instability during training if an even distribution of MEAN is adopted, but residues right before or right after CDRs can be missing. Here, we propose another way to initialize the residue positions in CDRs. First, we follow MEAN and select the residue right before and right after CDRs. If both residues exist, we take the mean coordinates. Otherwise, we use only the existing one. After that, we add some little noise like HERN to separate nodes and introduce some randomization to prevent overfitting, which is proven to bring slight improvements in performance.

**Loss Function.** Two sorts of losses are used for supervision. First, a cross-entropy (CE) loss is employed for sequence prediction as $\mathcal{L}_{seq} = \frac{1}{|\mathcal{V}_{HC}|} \sum_{v_i \in \mathcal{V}_{HC}} \text{CE}(p_i, c_i)$, where $c_i$ is the ground truth residue type for each node. Apart from that, a common RMSD loss is utilized for structure prediction and we leverage the Huber loss [37] to avoid numerical instability as $\mathcal{L}_{struct} = \text{Huber}\left(\mathbf{x}_{HC}^{(L)}, \mathbf{x}_{HC}\right)$, where the latter is the ground truth coordinates of the target CDR. The total loss is a weighted sum of the above two as $\mathcal{L} = \mathcal{L}_{seq} + \lambda \mathcal{L}_{struct}$, where $\lambda > 0$ is the balance hyperparameter.

## 3 Experiments

We assess our HTP via two mainstream challenging tasks: sequence-structure co-design in Section 3.1, and antibody sequence design based on antibody backbones in Section 3.2. Our evaluation is conducted in the standard SAbDab database [16]. More experimental setting details and data descriptions are elucidated in Appendix A.

### 3.1 Sequence-structure Co-design

**Task and Metrics.** In this task, we remove the original CDR from the antibody-antigen complex in the test set and try to co-design both sequence and structure of the removed region. Here we set the length of the CDR to be identical to the length of the original CDR. But in practice, the lengths of CDRs can be variable. For quantitative evaluation, we adopt amino acid recovery (AAR) and root-mean-squared error (RMSD) regarding the 3D predicted structure of CDRs as the metric. AAR is defined as the overlapping rate between the predicted 1D sequences and the ground truths. We also take advantage of TM score [38] to calculate the global similarity between the predicted and ground truth antibody structures. It ranges from 0 to 1 and evaluates how well the CDRs fit into the frameworks.

Table 2: Results of the fix-backbone design task on SAbDab.

| Model | CDR-H1 | | CDR-H2 | | CDR-H3 | |
|---|---|---|---|---|---|---|
| | AAR (%) ↑ | Perplexity ↓ | AAR (%) ↑ | Perplexity ↓ | AAR (%) ↑ | Perplexity ↓ |
| RosettaFix | $36.29 \pm 0.2$ | $14.78 \pm 0.01$ | $37.70 \pm 0.3$ | $12.30 \pm 0.02$ | $28.13 \pm 0.1$ | $24.05 \pm 0.14$ |
| Structured-TF | $53.24 \pm 3.2$ | $8.61 \pm 0.08$ | $49.87 \pm 1.2$ | $10.27 \pm 0.06$ | $30.29 \pm 0.4$ | $19.65 \pm 0.11$ |
| DiffAb | $59.91 \pm 1.2$ | $6.44 \pm 0.05$ | $59.14 \pm 1.8$ | $6.92 \pm 0.08$ | $33.30 \pm 0.5$ | $16.84 \pm 0.12$ |
| GVP-GNN | $62.72 \pm 1.5$ | $4.08 \pm 0.03$ | $62.48 \pm 1.7$ | $4.77 \pm 0.09$ | $34.59 \pm 0.6$ | $15.79 \pm 0.13$ |
| HTP | $\mathbf{86.01 \pm 1.1}$ | $\mathbf{1.71 \pm 0.04}$ | $\mathbf{64.46 \pm 1.3}$ | $\mathbf{3.29 \pm 0.06}$ | $\mathbf{43.25 \pm 1.2}$ | $\mathbf{9.01 \pm 0.10}$ |

| Model | CDR-L1 | | CDR-L2 | | CDR-L3 | |
|---|---|---|---|---|---|---|
| | AAR (%) ↑ | Perplexity ↓ | AAR (%) ↑ | Perplexity ↓ | AAR (%) ↑ | Perplexity ↓ |
| RosettaFix | $35.42 \pm 0.3$ | $15.82 \pm 0.01$ | $36.76 \pm 0.2$ | $14.67 \pm 0.01$ | $32.17 \pm 0.1$ | $18.01 \pm 0.00$ |
| Structured-TF | $56.73 \pm 3.1$ | $7.63 \pm 0.10$ | $52.11 \pm 1.8$ | $8.93 \pm 0.08$ | $43.48 \pm 0.7$ | $13.88 \pm 0.02$ |
| DiffAb | $58.82 \pm 1.6$ | $6.89 \pm 0.06$ | $55.40 \pm 1.2$ | $7.16 \pm 0.05$ | $47.31 \pm 0.5$ | $10.60 \pm 0.02$ |
| GVP-GNN | $60.18 \pm 1.4$ | $5.48 \pm 0.05$ | $59.66 \pm 1.5$ | $6.48 \pm 0.06$ | $51.34 \pm 0.6$ | $8.27 \pm 0.03$ |
| HTP | $\mathbf{93.62 \pm 1.5}$ | $\mathbf{1.09 \pm 0.08}$ | $\mathbf{91.46 \pm 1.7}$ | $\mathbf{1.58 \pm 0.10}$ | $\mathbf{80.71 \pm 1.1}$ | $\mathbf{2.66 \pm 0.10}$ |

**Baselines.** We pick up a broad range of existing mechanisms for comparison. Rosetta Antibody Design (**RAbD**) [9] is an antibody design software based on Rosetta energy functions. **RefineGNN** [10] is an auto-regressive model that first consider 3D geometry for antibody design and is E(3)-invariant. However, its original version is merely conditioned on the framework region rather than the antigen and the remaining part of the antibody. We follow Jin et al. [11] and replace its encoder with a message-passing neural network (MPNN) encoder and use the attention layer to extract information from the antibody-antigen representation. We name this modified model **C-RGNN** to make a distinction from its original architecture. Hierarchical Equivariant Refinement (**HERN**) [11] is the abbreviation of Hierarchical Equivariant Refinement Network, which employs a hierarchical MPNN to predict atomic forces and refine a binding complex in an iterative and equivariant manner. Its autoregressive decoder progressively docks generated antibodies and builds a geometric representation of the binding interface to guide the next residue choice. Multichannel Equivariant Attention Network (**MEAN**) [14] adopts a multi-round progressive full-shot scheme instead of an autoregressive one to output both 1D sequences and 3D structures. **DiffAb** [12] is a diffusion-based mechanism that achieves state-of-the-art performance on antibody design recently. It consists of three diffusion processes for amino acid types, coordinates, and orientations, respectively.

**Results and Analysis.** We run each model three times with different random seeds and report the mean and standard deviation of each metric in Table 1, where metrics are labeled with ↑/↓ if higher/lower is better, respectively. It can be found that our model outperforms all baselines by a significant margin in terms of AAR, RMSD, and TM-Score. Specifically, HTP brings an improvement of 53. 97%, 47. 42% and 51. 55% in AAR and 67. 54%, 57. 25% and 29. 75% in RMSD over the state-of-the-art DiffAb in H1, H2 and H3, respectively. This implies that our HTP might have a higher success rate in designing new antibodies targeting the given antigen. Moreover, it can also be expected that the performance of our HTP can be further improved as the number of antibodies and antigens with solved 3D structures keeps increasing. In addition, the light chain generally has a higher AAR and a lower RMSD than the heavy chain. For example, our model achieves nearly 90% AAR in CDR-L1 and CDR-L2. This phenomenon is consistent with the fact that CDR in the heavy chain is much longer and more variant.

## 3.2 Fix-backbone Sequence Design

**Task and Metrics.** This problem is more straightforward than the previous co-design task. The backbone structure of CDRs is given and only requires the CDR sequence's design. The fixed-backbone design is a common setting in protein design [39, 40] with an alias of inverse folding. We rely on the metrics of AAR introduced in Section 3.1 to examine the generated antibodies. The metric RSMD is ruled out since the backbone structures are fixed. We also rely on perplexity (PPL) to understand the uncertainty of model predictions, which is common for evaluating language models in natural language processing. In short, the perplexity calculates the cross entropy loss and takes its exponent.

Table 3: Comparison with pretrained antibody-specific language models.

| PLMs | CDR-H3 | | |
| --- | --- | --- | --- |
| | AAR (%) ↑ | RMSD ↓ | TM-Score ↑ |
| – | 25.31 ± 0.7 | 2.95 ± 0.02 | 0.9391 ± 0.005 |
| AbLang | 28.46 ± 1.6 | 2.88 ± 0.17 | 0.9435 ± 0.009 |
| AntiBERTa | 34.52 ± 1.2 | 2.41 ± 0.13 | 0.9473 ± 0.006 |
| HTP | **40.98 ± 1.5** | **2.06 ± 0.03** | **0.9621 ± 0.005** |

**Baselines.** Apart from some baselines that have been listed in the sequence-structure co-design problem, we compare HTP with three additional approaches. **RosettaFix** [41] is a Rosetta-based software for computational protein sequence design. **Structured Transformer** [42] (Structured-TF) is an auto-regression generative model that is able to sample CDR sequence given the backbone structure. **GVP-GNN** [43] extends standard dense layers to operate on collections of Euclidean vectors and performs geometric and relational reasoning on efficient representations of macromolecules.

**Results.** Table 2 documents the result. It is clearly found that our model achieves the highest AAR and the lowest PPL compared to all the baseline algorithms. To be specific, HTP leads to an increase of 37.13%, 3.01%, and 18.47% in AAR over the best GVP-GNN in H1, H2, and H3, separately, and 28.76%, 24.28%, and 49.90% in L1, L2, and L3, respectively. This demonstrates that our model is also effective in capturing the conditional probability of sequences given backbone structures. Furthermore, we can also observe that CDR-H3 is the hardest segment in comparison to other regions as its AAR is usually lower than 50%. Meanwhile, the average AAR of the light chain is more than 75%, indicating that the light chain maintains less diversity than the heavy chain.

### 3.3 Discussion

**Comparison with Antibody-specific Language Models.** Recently, emerging efforts have been paid to train large antibody-specific language models. Studies have also demonstrated that these models can capture biologically relevant information and be generalized to various antibody-related applications, such as paratope position prediction. Here, we make a further investigation into the efficacy of these antibody-specific language models for sequence-structure co-design. To be precise, we abandon the pretraining stages of the single-protein and antibody sequence levels and directly leverage external antibody-specific language models. As shown in Table 3, AntiBERTa and AbLang provide biologically relevant information that is beneficial for the challenge of co-design with an increase of 12.44% and 36.38% in AAR. However, their improvements are much smaller than those of PLMs trained through the first two levels of tasks in HTP.

**Up- and Downstream Protein.** Recently, Wang et al. [44] proposed a joint sequence-structure recovery method based on RosettaFold to scaffold functional sites of proteins. They perform the inpainting and fix-backbone sequence design tasks without immediate up- and downstream protein visible. However, we discover that our HTP can generate adequate diversity without the need to mask neighboring residues, *, that is,*, up- and downstream proteins. This difference stems from the fact that our HTP considers the entire antigen and available antibody as the context to complete the masked CDR rather than only depending on the tiny context of up- and downstream protein.

**Data Leakage of Language Models.** It is undisputed that the evaluation of design methods that use PLMs should be stringent and ensure that the test data were not previously seen by those pretrained models. However, ESM-2 is trained on all protein sequences in the UniRef database (September 2021 version), while our test set includes sequence released after December 2021, as well as structures with any CDR similar to those released after this date (with sequence identity higher than 50%). It is fairly possible that the training set of ESM-2 includes antibody sequences similar to the test set, leading to the intolerable data leakage problem.

Here, we conducted additional experiments aimed at assessing the contribution of ESM-2 in directly recovering CDR sequences. Specifically, we abandon the structural information and re-generate CDRs entirely based on sequential information. Towards this end, we first extract residue-level representations via (fixed-weight) ESM-2 and feed them to a three-layer perceptron to predict the masked CDR-H3, where no antigen sequences are given. The results show that this algorithm only

achieved an AAR of 14.63% to recover CDR-H3, much lower than all baseline methods such as RAdD (21.73%) and our HTP (40.98%). This compellingly demonstrates that ESM-2 is not the primary driver of our favorable numerical outcomes and the experimental benefits brought forth by HTP are not due to data leakage. This also accords with ATUE's [45] findings that ESM models perform well in low-antibody-specificity-related tasks but can even bring negative impacts for high-antibody-specificity-related tasks. We have also provided adequate evidence in Appendix 4 to show that the other pretraining resources are free from any possible data leakage concern.

## 4    Conclusion

Antibodies are crucial immune proteins produced during an immune response to identify and neutralize the pathogen. Recently, several machine learning-based algorithms have been proposed to simultaneously design the sequences and structures of antibodies conditioned on the 3D structure of the antigen. This paper introduces a novel approach called the hierarchical training paradigm (HTP) to address the co-design problem. It leverages both geometric neural networks and large-scale protein language models and proposes four levels of training stages to efficiently exploit the evolutionary information encoded in the abundant protein sequences and complex binding structures. Extensive experiments confidently show that each stage of HTP significantly contributes to the improvements of the deep learning model's capacity in predicting more accurate antibody sequences and storing its corresponding 3D structures. Instead of focusing on the architecture side, our study hopes to shed light on how to better blend protein data of different modalities (*i.e.*, one- and three-dimensions) and domains (*i.e.*, antibodies, and non-antibodies) for tackling the sequence-structure co-design challenge, which has been long ignored by existing works.

## 5    Limitations and Future Work

In spite of the promising progress of our HTP, there is still some space left for future explorations. First, more abundant databases can be exploited in our framework. For example, AntiBodies Chemically Defined (ABCD) [46] is a large antibody sequence database that can be used to improve the capacity of protein language models at the second level. We do not use it in our work because our request for this database has not been approved by the authors so far. Secondly, we fix the language models during the last two levels of training (*i.e.*, levels that need complex structure prediction) for simplicity and use them as the node feature initializer. It might be beneficial if both the PLM and the geometric encoder are tuned.

### Acknowledgments and Disclosure of Funding

This work was supported by National Key R&D Program of China (No. 2022ZD0115100), National Natural Science Foundation of China Project (No. U21A20427), and Project (No. WU2022A009) from the Center of Synthetic Biology and Integrated Bioengineering of Westlake University. F.W. and S.L. led the research. F.W. contributed technical ideas and developed the method and performed analytics. S.L. provided evaluation and suggestions. The authors thank Professor Siqi Sun and Tao You from Shanghai Artificial Intelligence Lab (SAIL) for their efforts in processing the OAS data, and Professor Buyong Ma from Shanghai Jiaotong University for his comments on improving the quality of the paper.

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

# Appendix

## A   Experimental setting

**Data Descriptions for Different Stages of HTP.**   (1) For the single-protein sequence level, we employ the state-of-the-art ESM-2 [47], which outperforms all tested single-sequence protein language models across a wide range of structure prediction tasks and enables prediction of the atomic resolution structure. It is trained on 86 billion amino acids across 250 million protein sequences that span evolutionary diversity. Specifically, ESM uses UniRef50, September 2021 version. The training dataset was partitioned by randomly selecting 0.5% ($\approx$ 250,000) sequences to form the validation set. The training set has sequences removed via the procedure described in Hie et al. [48]. ESM-2 runs the MMseqs search to obtain the query and target databases. All train sequences that match a validation sequence with 50% sequence identity in this search are removed from the train set. The details of the ESM series can be found in `https://github.com/facebookresearch/esm`.

(2) For the antibody sequence level, we use the Observed Antibody Space database (OAS) [31] and its subsequent update [32] as the pretraining data. It currently contains more than one billion sequences, from more than 80 different studies that cover various immune states, organisms, and individuals, which can be downloaded from its official website at `https://opig.stats.ox.ac.uk/webapps/oas/`. We upload the processed paired data in `https://pan.baidu.com/s/18lB8gl9Maf0nnNPIw83ZzA?pwd=1212` (password: 1212) as well as the unpaired data in `https://pan.baidu.com/s/161gU8fso6rz6-QGfNoCoHQ?pwd=96uF` (password: 96uf).

(3) For the protein-protein complex structure level, we use the Database of Interacting Protein Structures (DIPS) [35]. It is a larger protein complex structure dataset than existing antibody-antigen complex structure datasets and is extracted from the Protein Data Bank [36]. We attain the database from Atom3d in Zendo `https://zenodo.org/record/4911102`, which is a collection of both novel and existing benchmark datasets spanning several key classes of biomolecules. Referring to Atom3d, we split protein complexes by sequence identity at 30%, resulting in train/validation/test sets with 87,303/31,050/15,268 instances.

(4) For the antibody-antigen complex structure level, we select all available antibody-antigen protein complexes from SAbDab [16] at `https://opig.stats.ox.ac.uk/webapps/newsabdab/sabdab/`, leading to a dataset containing 9,823 structures. CDRs are identified using the antibody numbering program AbRSA [49]. Following the setting in [12], the chosen data points are divided into training and test data based on their release date and CDR sequence identity. To be explicit, the test split contains protein structures released after December 24, 2021, as well as structures with any CDR similar to those released after this date with a sequence identity higher than 50%. The antibodies in the test set are further clustered with a CDR sequence identity of 50% to eliminate duplicates, resulting in 20 antibody-antigen structures. The training and validation splits just include complexes not involved during the curation of the test split. After that, we randomly divided the remaining complexes with a ratio of 90% and 10% into training and validation sets.

**Dataset Sequence Similarity.** In our splitting strategy, the test set of SAbDab includes sequence released after December 2021, as well as structures with any CDR similar to those released after this date (with sequence identity higher than 50%). It is fairly possible that the pretraining datasets include antibody sequences similar to the test set. To navigate this issue, we have conducted a comprehensive analysis of the sequence similarity between the various pretraining data sources and the test set in SAbDab. This analysis includes general protein sequences from general protein-protein complexes from DIPS and antibodies from OAS. We have plotted the sequence similarity distributions in Figure 4 and present the statistical findings of this analysis in Table 4.

Table 4: The statistics of the sequence similarity between our splitted SAbDab test set and different pretraining datasets.

| Dataset | Mean | Std. | Min. | 25% | 50% | 75% | Max. |
|---------|-------|-------|-------|-------|-------|-------|-------|
| DIPS | 0.188 | 0.036 | 0.000 | 0.183 | 0.198 | 0.208 | 0.429 |
| OAS | 0.246 | 0.017 | 0.200 | 0.235 | 0.243 | 0.254 | 0.401 |

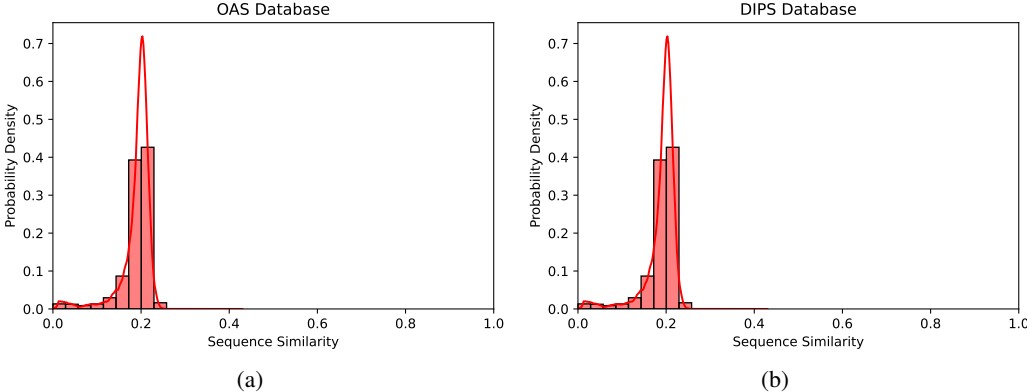

Figure 4: The distributional plot of sequence similarity between different pre-training resources and the test set in SAbDab.

The statistical findings unequivocally demonstrate that the highest sequence similarity values across these three datasets are consistently below 0.5. This empirical evidence serves as a strong basis for our resolute assertion that neither DIPS, nor OAS encompasses sequences that exhibit significant similarity to the SAbDab test set. With this robust evidence at hand, we maintain our firm confidence that the hierarchical training paradigm we have employed is free from any potential data leakage concerns.

**Implementation Details.** HPT is implemented in PyTorch and PyTorch Geometric packages. For all four training stages, we leverage an Adam optimizer [50] with a weight decay of 1e-5. All experiments are run on multiple A100 GPUs, each with a memory storage of 80G.

(1) For ESM-2 in the single-protein sequence level training, we adopt a middle-size version, which has a parameter number of 150M, 30 layers, and a hidden dimension of 640. Besides, we append the ESM-2 with a three-layer perceptron to forecast the residue type for MLM.

(2) For the antibody sequence level training, we use a batch size of 2 to avoid out-of-memory error and 4 workers to load the data. The number of epochs is 100 and starting learning rate is 1e-5. Apart from that, we utilize a ReduceLROnPlateau scheduler with a factor of 0.6, patience of 5 epochs, and a minimum learning rate of 1e-7.

(3) For protein-protein complex structure level training, we use a batch size of 32, 1000 epochs, and 4 works to speed up data loading. The starting learning rate is 1e-4, and a ReduceLROnPlateau scheduler is utilized to automatically adjust the learning rate with a factor of 0.6 and patience of 3 epochs. We adopt a distance threshold of 8.0Åto determine the connection between different graph nodes (*i.e.*, the alpha carbon of each residue). As for the loss weight balance, we set $\lambda = 1$.

(4) For the antibody-antigen complex structure level training, we also adopt the distance threshold of 8.0Åto build the graph connection. For the random initialization of the CDR coordinates, we use a noise of $\epsilon = 0.1$. As for the other important hyperparameters, we use a grid search mechanism to find the optimal combination. Notably, the geometric neural networks used in the third and fourth levels are matched to each other. If we alter the setting of GGNNs in the antibody-antigen complex structure level training, we need to retrain it in the protein-protein complex structure level first. The entire hyperparameter search space is depicted in Table 5.

**Reproduction of Baselines.** Concerning the implementation of several baseline methods, we use the official repositories for conditional RefineGNN ( `https://github.com/wengong-jin/RefineGNN/`), HERN ( `https://github.com/wengong-jin/abdockgen`), DiffAb ( `https://github.com/luost26/diffab`). To reproduce the performance of existing antibody-specific pretrained PLMs, we download the code from `https://github.com/alchemab/antiberta` for AntiBERTa and `https://github.com/oxpig/AbLang` for AbLang. In our comparison, we directly use their pretrained residue features as the input for GGNNs without any fine-tuning.

Table 5: Hyperparameters setup for HTP.

| Hyperparameters Search Space | Symbol | Value |
|---|---|---|
| **Training Setup** | | |
| Epochs | – | [100, 500, 1000] |
| Batch size | – | [32, 64, 128] |
| Learning rate | – | [1e-4, 5e-5, 1e-6, 1e-7] |
| Warmup | – | [Yes, No] |
| Warmup epochs | – | [10, 20] |
| Loss Balance weight for Coordinates and Residue Types | $\lambda$ | [0.1, 0.3, 0.5, 0.7] |
| **GNN Architecture** | | |
| Dropout rate | – | [0.1, 0.2] |
| Number of GNN layers | $L$ | [2, 4, 6] |
| Tanh activation function | – | [Yes, No] |
| Coordinate Normalization | – | [Yes, No] |
| The hidden dimension of node representations | – | [320, 640] |
| The hidden dimension of edge representations | – | [16, 32, 64] |

**Code Availability.** All relevant Python code to reproduce the results in our paper is stored in the GitHub repository at `https://github.com/smiles724/HTP`.

# B Additional Results

## B.1 Ablation Study

We investigate the effectiveness and necessity of each component of our HTP. As shown in Table 6, the elimination of the level of protein-protein complex structure level level induces performance degradation, where RMSD increases from 2.06 to 2.49. Moreover, we implement a variant of HTP by replacing features obtained by pretrained PLMs with learnable embedding features, whose performance is worse than HTP. To be concise, AAR declines from 40.98 to 25.31, and RMSD increases from 2.06 to 2.65. In summary, our HTP brings significant relative improvements of 78.56% in AAR, 41.97% in RMSD, and 2.94% in TM-Score. This phenomenon strongly supports the superiority of our approach over existing naive co-design algorithms that are trained only on antibody-specific structure data.

Table 6: Effects of each module, where SPS stands for the single-protein sequence level, PPCS denotes the protein-protein complex structure level, and AS represents the antibody sequence level. The last row computes the relative improvements of HTP over the primitive baseline without any protein data augmentation.

| | SPS | AS | PPCS | SAbDab (CDR-H3) | | |
|---|---|---|---|---|---|---|
| | | | | AAR (%) ↑ | RMSD ↓ | TM-Score |
| 1 | ✗ | ✗ | ✗ | $22.95 \pm 0.5$ | $3.55 \pm 0.01$ | $0.9146 \pm 0.003$ |
| 2 | ✓ | ✗ | ✗ | $33.87 \pm 0.8$ | $2.77 \pm 0.04$ | $0.9450 \pm 0.006$ |
| 3 | ✓ | ✓ | ✗ | $\underline{38.42 \pm 1.6}$ | $\underline{2.49 \pm 0.03}$ | $\underline{0.9538 \pm 0.004}$ |
| 4 | ✗ | ✗ | ✓ | $25.31 \pm 0.7$ | $2.95 \pm 0.02$ | $0.9391 \pm 0.005$ |
| 5 | ✓ | ✓ | ✓ | $\mathbf{40.98 \pm 1.5}$ | $\mathbf{2.06 \pm 0.03}$ | $\mathbf{0.9621 \pm 0.005}$ |
| Imp. | – | – | – | 78.56% | 41.97% | 2.94% |

# C Related Work

**Antibody Design.** The majority of old-school computational approaches for antibody design are based on sampling algorithms over handcrafted and statistical energy functions to iteratively modify protein sequences and structures [8, 9]. These physics-based algorithms are computationally expensive and prone to be stuck in local energy minimum, which triggers the adaptation of deep

learning in this sub-field. The initial researchers [4, 5] use pure PLMs to generate protein sequences but disregard the available antigen structures.

To circumvent this, Jin et al. [10] introduce RefineGNN, the first co-design architecture that aims to neutralize SARS-CoV-2. Later, HERN [11] is proposed as a more general version for paratope docking and design, which opens the door to produce antibodies given arbitrary antigen structures. Subsequent efforts are spent in either modifying the generative style or utilizing more advanced deep learning architectures such as diffusion denoise probabilistic models (DDPMs). For example, DiffAb [12] achieves atomic-resolution antibody design with SO(3)-equivariance, while MEAN [14] corrects the autoregressive manner with a full-shot one to prevent low efficiency and accumulated errors during inference.

**Protein Sequence Modeling.**    Sequence-based protein representation learning is mainly inspired by the field of natural language processing. A large body of early work is focused on modeling individual protein families [51], solving problems such as functional nanobody design [5]. The success of this method then motivates the prospective trend to model large-scale databases of protein sequences by means of unsupervised learning. This line of study targets capturing the biochemical and co-evolutionary knowledge that underlies a large-scale protein sequence corpus by self-supervised pertaining. Thanks to them, a number of pertaining objects have been explored such as the next amino acid prediction [4, 24], masked language modeling (MLM) [51, 22], pairwise MLM [52], contrastive predictive coding [53], conditional generation [54], and position-specific scoring matrix prediction [55]. In addition, another line [56, 57] is based on multiple sequence alignment (MSA), leveraging sequences within a protein family to seize the conserved and variable regions of homologous sequences. Notably, some schemes for protein sequence modeling also seek to incorporate structural information in either the pretraining stage [58] or the finetuning stage [59].

The improvements in model scale and architecture are also crucial to the recent achievement of PLMs. Explicitly, Rao et al. [51] evaluate various PLMs in a panel of benchmarks and discover that multi-head attention outpaces the Potts model in contact prediction, even if using a single sequence for inference. Concurrently, Vig et al. [60] observe that specific attention heads of pretrained Transformers have straight correlations with protein contact. Others [24] investigate a variety of Transformer variants and demonstrate that large Transformers can obtain state-of-the-art features in various tasks. Apart from that, the latest ESM-2 [47] trains the largest PLM with 15B parameters and shows that as models are scaled, they learn information enabling the protein structure prediction at the resolution of individual atoms.

**Protein Structure Learning.**    With the rapid advance of geometric deep learning, it has become increasingly attractive and challenging to represent and reason about the structures of macromolecules in 3D space. For the sake of encoding spatial information in protein structures including bond lengths and dihedral angles, numerous 3D geometric neural networks such as 3DCNN [61] or GNNs [42, 43, 62, 63] have been invented. They excel at capturing complex interactions between sets of amino acids [64] and attain pivotal Euclidean geometry, *e.g.*, E(3) or SE(3)-equivariance and symmetry.

However, compared to protein sequences in databases like UniProt [65] or Pfam [66], the known structures in the PDB are scarce and hard to obtain. Therefore, it becomes urgent to develop structure-based mechanisms to efficiently learn protein representations with much fewer pretraining data. For instance, Hermosilla and Ropinski [17] use contrastive learning in terms of molecular substructures to help models understand protein structure similarity and functionality. Moreover, Chen et al. [67] propose a self-supervised framework that predicts angles and inter-residue distances. Additionally, Guo et al. [68] present a coordinate denoising score matching method. Wu et al. [21] put forward a novel prompt-based denoising conformation generative pretraining method based on the trajectories of molecular dynamics simulations. A recent attempt [34] makes a combination of both contrastive learning and self-prediction with more intriguing augmentation functions. Despite this progress, all of them are dealing with single-protein structures. No preceding studies have considered structure-based pretraining in the circumstance of multiple proteins. That is, how to pretrain on protein-protein complex, or more specifically, the antibody-antigen complex, remains unexplored.

