# OpenReview forum: "A Hierarchical Training Paradigm for Antibody Structure-sequence Co-design"
_NeurIPS.cc/2023/Conference — NeurIPS 2023 poster_

### Official Review · Reviewer_eNMD · 2023-06-17

**Soundness:** 3 good
**Presentation:** 3 good
**Contribution:** 3 good
**Rating:** 6
**Confidence:** 4

**Summary:**

This paper introduces a novel approach called the hierarchical training paradigm (HTP) to address the antibody co-design problem. It leverages both geometric neural networks and large-scale protein language models and proposes four levels of training stages to efficiently exploit the evolutionary information encoded in the abundant protein sequences and complex binding structures.

**Strengths:**

1. The paper is well-written and easy to follow.
2. The authors explored using multi-modal data from different domains to enhance the antibody sequence-structure co-design performance, which is a novel perspective.
3. Extensive experiments show that HTP significantly overperforms previous methods.

**Weaknesses:**

1.  The code is not provided at the current stage.
2.  The reported baseline results are inconsistent with the original papers. For example, in table. 1, the CDR-H3 AAR of MEAN is only 22.56%, which is much lower than the 36.38% in the original paper. Therefore, the comparison may be unfair.

**Questions:**

Please see the weakness.

**Limitations:**

The authors have adequately discussed the limitations.

---

> ### Author Rebuttal · Authors · 2023-08-02
>
> We would like to express our gratitude for your valuable feedback and constructive comments on our paper. We appreciate your recognition of the strengths of our work and acknowledge the weaknesses pointed out. We have carefully reviewed your feedback and would like to address each concern:
>
> 1. Code Availability: We apologize for not providing the code at the current stage. We fully understand the importance of code reproducibility for advancing research in the field and will release the code and associated resources upon acceptance.
>
> 2. Inconsistent Baseline Results: We appreciate your observation regarding the inconsistency of the reported baseline results with the original papers. It is worth noting that we follow DiffAb [A] and adopt a completely different dataset split from MEAN [B]. To be specific, the selected data points in SAbDab are divided into training and test data based on their release date and CDR sequence identity. The test split includes protein structures released after December 24, 2021, and structures with any CDR similar to those released after the date (sequence identity higher than 50%). Antibodies in the test set are further clustered with 50% CDR sequence identity to remove duplicates, finally resulting in 20 antibody-antigen structures. The training split contains complexes not involved during the curation of the test split. Meanwhile, MEAN [B] uses the split setting of RefineGNN [C], which separates the entire dataset into training, validation, and test sets according to the clustering of CDRs to maintain the generalization test. Then they divide all clusters into training, validation, and test sets with a ratio of 8:1:1.
> As a consequence, the split setting of DiffAb [A] poses a greater challenge than that of MEAN [B] and RefineGNN [C]. This statement can be verified via the discrepancy in their reported results. For instance, RefineGNN [C] achieves an AAR of 39.40%, 37.06%, and 18.88% for CDR-H1, CDR-H2, and CDR-H3 in the MEAN paper [B], but only reaches an AAR of 27.77%, 27.04%, and 8.00% for CDR-H1, CDR-H2, and CDR-H3 in the DiffAb paper [A]. This phenomenon indicates that the same algorithm can encounter a decline in its performance when facing a more difficult data-splitting mechanism. Based on this fact, we believe our reproduction of several important baselines is implemented well and the comparison in our manuscript is fair. However, we are still thankful for your concerns and will add a few sentences to clarify the influence of dataset splitting over the model performance.
>
>
> [A] Luo, Shitong, et al. "Antigen-specific antibody design and optimization with diffusion-based generative models for protein structures." Advances in Neural Information Processing Systems 35 (2022): 9754-9767.
>
> [B] Kong, Xiangzhe, Wenbing Huang, and Yang Liu. "Conditional antibody design as 3d equivariant graph translation." arXiv preprint arXiv:2208.06073 (2022).
>
> [C] Jin, Wengong, et al. "Iterative refinement graph neural network for antibody sequence-structure co-design." arXiv preprint arXiv:2110.04624 (2021).

---

> > ### Comment · Reviewer_eNMD · 2023-08-16
> > **Reply to the authors**
> >
> > I have read the reply and appreciate the author's reply. My concerns are mostly resolved. Thanks!

---

### Official Review · Reviewer_H5io · 2023-06-21

**Soundness:** 3 good
**Presentation:** 4 excellent
**Contribution:** 4 excellent
**Rating:** 5
**Confidence:** 4

**Summary:**

In this study, the author points out that the efficacy of existing co-design methods is predominantly limited by the small number of antibody structures. They propose a hierarchical training paradigm (HTP), a novel unified prototype to exploit multiple biological data resources and aim at fully releasing the potential of geometric graph neural networks (GGNNs) for the sequence and structure co-design problem.

**Strengths:**

- This study is well-motivated, and sheds light on integrating big data (including sequence data, protein complex data) into antibody design.
- The writing is clear and the pipeline is easy to follow.

**Weaknesses:**

- Missing comparison with the SOTA method, dyMEAN, whose AAR of CDR-H3 is 43.65% (Table 4). (End-to-End Full-Atom Antibody Design, ICML 2023)
- Some results are inconsistent with the reference. In the original paper of MEAN, the AAR of CDR-H3 is reported to be 36.38% or 39.87% under different evaluation settings (Table 1, MEAN), while in this manuscript, MEAN's AAR is 22.56% (Table 1).
- Some results are confusing. The performance on fix-backbone design (Table 2) is lower than that on co-design (Table 1). For example, the AARs of CDR-L1/L2/L3 on co-design are 91.13%/89.80%/73.82%, while the performance on fix-backbone design is only 77.49%/74.15%/76.96%. The AARs of CDR-H3 are both 40.98% on two tasks.  Fixed-backbone design should be much easier than the co-design task. Can you give an explanation?

**Questions:**

- In line 590, Appendix, is weight decay 1e-5?
- Missing References: Wang et al, On Pre-training Language Model for Antibody, ICLR 2023
- How do you split the epitope from the antigen? Is there any threshold?
- In this work, you build a graph with a cutoff of 8A, and initialize the antibody with the center of the residues before/after the CDR. Is it possible that the initialized antibody is far from the antigen, i.e., larger than 8A, such that the antibody can not interact with the antigen?

---

> ### Author Rebuttal · Authors · 2023-08-02
>
> Thank you for your thorough review of our study. We appreciate your positive feedback on the motivation of our work and the clarity of the writing, as well as your constructive comments that will help us improve the quality and accuracy of the paper. We have carefully considered each of the points you raised and would like to address them accordingly:
>
> * Comparison with dyMEAN
>
> dyMEAN [A], indeed, is an extraordinary work for antibody sequence-structure co-design and extends the framework of MEAN [B]. However, the problem setting of dyMEAN [A] is completely different from most existing co-design architectures (i.e., MEAN [B], RefineGNN [C], HERN [D], DiffAb [E], and ours). To be explicit, dyMEAN [A] assumes that the 1D incomplete antibody sequences (without CDRs) are given but no antibody structures are provided. To resolve this dilemma, dyMEAN proposes an end-to-end pipeline to replace the previous multi-stage schema: IgFold for structure prediction + HDock for docking on the target epitope + MEAN [B] for binding CDR generation. In contrast, we follow the conventional setting of RefineGNN [C], HERN [D], DiffAb [E], and MEAN [B], and assume that the incomplete antibody structures (without CDRs) are offered. Therefore, it is hard to directly compare the effectiveness of those co-deign approaches with dyMEAN [A]. In Table 1 of the dyMEAN paper [A], the reported results for DiffAb [E], MEAN [B], and HERN [D] all adopt the above-mentioned pipeline (IgFold -> HDock -> CDR generation -> Rosetta) rather than only employing those algorithms. Thus, we are unable to fully compare dyMEAN in our experimental section.
>
>
> * Inconsistent Results of MEAN
>
> Thank you for bringing to our attention the discrepancy in the reported baseline results compared to the original papers. We acknowledge the importance of this observation and would like to clarify the differences in the dataset-splitting mechanism used in our work compared to previous studies. In our study, we followed the dataset-splitting approach of DiffAb [E], which uses a completely different methodology compared to MEAN [B]. Specifically, we utilized the SAbDab dataset and split the data points into training and test sets based on their release date and CDR sequence identity. The test set includes protein structures released after December 24, 2021 and structures with any CDR similar to those released after that date (sequence identity higher than 50%). To eliminate duplicates, we further clustered antibodies in the test set with 50% CDR sequence identity, resulting in a final set of 20 antibody-antigen structures. On the other hand, the training split contains complexes that were not involved in the curation of the test split. In contrast, MEAN [B] employs the split setting of RefineGNN [C], which divides the entire dataset into training, validation, and test sets based on the clustering of CDRs to maintain generalization. Subsequently, all clusters are divided into training, validation, and test sets in an 8:1:1 ratio.
>
> Due to the different data-splitting mechanisms, our approach in DiffAb [E] poses a greater challenge than that of MEAN [B] and RefineGNN [C]. This challenging dataset split is likely responsible for the observed discrepancies in the reported results. For instance, RefineGNN [C] achieves an AAR of 39.40%, 37.06%, and 18.88% for CDR-H1, CDR-H2, and CDR-H3 in the MEAN paper [B]. However, in the DiffAb paper [E], the AAR for CDR-H1, CDR-H2, and CDR-H3 drops to 27.77%, 27.04%, and 8.00%, respectively. This difference indicates that the same algorithm may experience a decline in performance when facing a more challenging data-splitting mechanism. In light of this fact, we believe that our reproduction of several important baselines is well-implemented, and the comparison in our manuscript is fair, considering the more difficult dataset splitting used in our work. However, we are grateful for your concerns, and we will add a few sentences to our paper to clarify the influence of dataset splitting on the model performance.
>
> * Performance Discrepancy in Fix-Backbone Design
>
> We appreciate your astute observation regarding the contrast in performance between the co-design and fix-backbone design tasks. Indeed, we concur with your insight that fix-backbone design should be relatively less complex since it doesn't entail predicting the precise positions of CDRs.
>
> In our initial implementation for CDR-L1/L2/L3, we admit that we did not strictly follow a grid search of the entire hyperparameter space. Instead, we opted for early stopping during training for convenience once we observed an acceptable result that outperformed all existing baselines. We acknowledge that this approach might not have fully explored the optimal combination of different hyperparameters. In response to your valuable feedback, we have re-run the experiments and conducted a thorough grid search to find the best combination of hyperparameters for CDR-L1/L2/L3 in fix-backbone design. The updated results are listed below, and as you can see, the overall performance of CDR-L1/L2/L3 in fix-backbone design is indeed better than that of the sequence-structure co-design task:
>
> | Region | CDR-L1 |  | CDR-L2 |  | CDR-L3    |    |   |
> |--|---|-----|---|----|---|--|---|
> |  Metric   | AAR               | Perplexity        | AAR               | Perplexity        | AAR               | Perplexity        |   |
> | HTP | 93.62$\pm$1.5 | 1.09$\pm$0.08 | 91.46$\pm$1.7 | 1.58$\pm$0.10 | 80.71$\pm$1.1 | 2.66$\pm$0.10 |   |
>
> Additionally, we extend our gratitude for identifying a typographical error in Table 2. To rectify this, we'd like to clarify that the AAR for CDR-H3 in the fix-backbone design task stands at 43.25. We sincerely appreciate your attention to detail and your insistence on rigor in our experimental setup. Your feedback has led us to reevaluate our approach and perform a comprehensive grid search, which has yielded more reliable and accurate results.

---

> > ### Author Response · Authors · 2023-08-11
> > **Rebuttal by Authors (Continued)**
> >
> > * Weight Decay in Line 590
> >
> > Yes, you are correct. The weight decay in line 590, as described in the appendix, is set to 1e-5. We will ensure that this information is clearly stated in the revised version.
> >
> > * Missing References
> >
> > We apologize for the oversight in missing the reference to EATLM [F], which is an excellent study introducing a new pre-trained
> > antibody language model EATLM. It leverages an ancestor germline prediction (AGP) task and a mutation position prediction (MPP) task to enable evolution awareness. However, it is a great pity that until we submitted our manuscript to NeurIPS, EATLM did release the code but did not make the pretrained weights publicly available. Actually, the pretrained language model weights are still not accessible today (see https://github.com/dqwang122/EATLM). Therefore, we fail to utilize EATLM and examine its benefits for our antibody design problem. As a remedy, we only compare our algorithm with some existing antibody-specific language models containing AntiBERTa [G] and AbLang [H]. But as you suggested, we will include some necessary discussion about EATLM in the revised version of the paper.
> >
> > * Epitope-Antigen Split
> >
> > The splitting of the epitope from the antigen is an essential step in our method. In HERN [D], they selected the $m\in [20, 40, 80]$ closest residues to the antibody as the epitope. However, as for our epitope-based CDR coordinate initialization, we adopt the widely-acknowledged way to determine the epitope residues based on their proximity to the antibody residues. That is, we recognize antigen residues as epitopes if their heavy atoms are within 8A of another heavy atom from any antibody chain (either heavy or light). This aligns with several important studies in epitope prediction [I]. We will provide additional details and clarify this threshold used for the epitope-antigen split in the revised manuscript.
> >
> > * Graph Initialization
> >
> > We acknowledge your concern about the possibility of initializing the antibody far from the antigen, potentially exceeding the predefined cutoff. To better understand the impact of graph connectivity between antigens and initialized CDRs, we compute the explicit distance between the initialized CDR and the antigen, and results show that 15.26% CDRs cannot interact with the antigen within a threshold of 8A. This phenomenon points out a possible direction to further promote the performance of our model by building interactions between the initialized CDR and the antigen. For instance, we can enlarge the receptor field of the initialized CDR from 8A to 12A or 16A. Or instead, we can connect the initialized CDR to the $k$ closest residues in the antigen.
> >
> > [A] Kong, Xiangzhe, Wenbing Huang, and Yang Liu. "End-to-End Full-Atom Antibody Design." arXiv preprint arXiv:2302.00203 (2023).
> >
> > [B] Kong, Xiangzhe, Wenbing Huang, and Yang Liu. "Conditional antibody design as 3d equivariant graph translation." arXiv preprint arXiv:2208.06073 (2022).
> >
> > [C] Jin, Wengong, et al. "Iterative refinement graph neural network for antibody sequence-structure co-design." arXiv preprint arXiv:2110.04624 (2021).
> >
> > [D] Jin, Wengong, Regina Barzilay, and Tommi Jaakkola. "Antibody-antigen docking and design via hierarchical structure refinement." International Conference on Machine Learning. PMLR, 2022.
> >
> > [E] Luo, Shitong, et al. "Antigen-specific antibody design and optimization with diffusion-based generative models for protein structures." Advances in Neural Information Processing Systems 35 (2022): 9754-9767.
> >
> > [F] Wang, Danqing, Y. E. Fei, and Hao Zhou. "On pre-training language model for antibody." The Eleventh International Conference on Learning Representations. 2022.
> >
> > [G] Olsen, Tobias H., Iain H. Moal, and Charlotte M. Deane. "AbLang: an antibody language model for completing antibody sequences." Bioinformatics Advances 2.1 (2022): vbac046.
> >
> > [H] Leem, Jinwoo, et al. "Deciphering the language of antibodies using self-supervised learning." Patterns 3.7 (2022).
> >
> > [I] Tubiana, Jérôme, Dina Schneidman-Duhovny, and Haim J. Wolfson. "ScanNet: an interpretable geometric deep learning model for structure-based protein binding site prediction." Nature Methods 19.6 (2022): 730-739.

---

> > > ### Comment · Reviewer_H5io · 2023-08-19
> > >
> > > I have read the reply and appreciate the author's effort to make the clarification. Thank you!

---

### Official Review · Reviewer_81S2 · 2023-07-06

**Soundness:** 1 poor
**Presentation:** 3 good
**Contribution:** 3 good
**Rating:** 5
**Confidence:** 3

**Summary:**

This paper proposes a hierarchical training paradigm for antibody sequence-structure codesign. It incorporates different sources of data, including general protein sequences, antibody sequences, general protein-protein complexes, and antibody-antigen complexes. The motivation is that there are a lot more data on general proteins and pre-training the model on these non-antibody data may provide additional boost to model performance. Specifically, it uses the pre-trained ESM-2 language model to incorporate all the knowledge learned from general protein sequence data. It then fine-tunes it on all antibody sequences in Observed Antibody Space (OAS). The fine-tuned language model is used to calculate features for antibody and antigen sequences. Next, the model is trained on all protein-protein complexes in DIPS. Lastly, the model is fine-tuned on antibody-antigen complexes in SAbDab. The method shows substantial improvement on antibody design benchmarks compared to existing baselines.

**Strengths:**

* This paper proposes to incorporate multiple sources of biological data for model pre-training. If implemented properly, this is a important contribution to the field.
* The evaluation is comprehensive. It includes all the recent baselines proposed in the field. The improvement over existing methods are substantial (but it can be due to potential data leakage, see discussion below).

**Weaknesses:**

* The main limitation of this paper is potential data leakage. ESM-2 is trained on all protein sequences in the UniRef database (September 2021 version). The test set includes sequence released after December 2021, as well as structures with any CDR similar to those released after this date (with sequence identity higher than 50%). Therefore, it is fairly possible that the training set of ESM-2 includes antibody sequences similar to the test set. Likewise, OAS may also contain antibody sequences similar to the test set and the author did not perform any filtering to ensure this may not happen. Similarly, DIPS also contains antibody-antigen structures. Even though it only contains structures released before 2019, it may still contain similar antibody sequences or antigen sequences to the test set. Can the authors provide evidence that there is no potential data leakage?
* The model architecture is an adaptation of existing models (e.g., EGNN). The technical innovation of the model architecture is relatively weak (though this is not the focus of the paper).
* The antibody-antigen complex test set contains only 21 structures, which is too small for evaluation. Expansion of the test set is necessary (at least over 100 structures is needed).

**Questions:**

* Can you plot sequence similarity between all the different training data sources (general protein sequences in UniRef50, general protein-protein complexes in DIPS, antibodies in OAS)?

**Limitations:**

The authors has addressed the limitations and negative societal impact in the appendix.

---

> ### Author Rebuttal · Authors · 2023-08-08
>
> Thank you for your valuable feedback and comments on our paper. We appreciate your acknowledgment of the potential importance of incorporating multiple sources of biological data for model pre-training. We have carefully considered your concerns and would like to address them as follows:
>
> * Data Leakage Or Not
>
> (1) We acknowledge the concern you raised regarding the potential high sequence similarity between the different datasets used in our proposed approach. In response, we have conducted a comprehensive analysis of the sequence similarity between the various pretraining data sources and the test set in SAbDab. This analysis includes general protein sequences from UniRef50, general protein-protein complexes from DIPS, and antibodies from OAS. We have plotted the sequence similarity distributions and have attached the corresponding figures in the uploaded PDF in the general author rebuttal. Below, we present the statistical findings from this analysis:
>
> |   Dataset           | Mean              | Std               | Min.              | 25%               | 50%               | 75% | Max. |
> |-----------------|-------------------|-------------------|-------------------|-------------------|-------------------|-----|------|
> | UniRef50 |  0.051 | 0.021 | 0.002 | 0.041 |0.052 | 0.059 | 0.260   |
> | DIPS | 0.188 | 0.036 | 0.000 | 0.183 | 0.198 | 0.208 | 0.429 |
> | OAS | 0.246 | 0.017 | 0.200 | 0.235 | 0.243 | 0.254 | 0.401   |
>
> From the statistical results, it is evident that the maximum sequence similarity values for these three datasets are all below 0.5. Based on this evidence, we can confidently conclude that neither UniRef50, DIPS, nor OAS contains sequences that are substantially similar to the SAbDab test set. Consequently, we firmly believe that there is no potential data leakage in our hierarchical training paradigm.
>
>
> (2) Furthermore, we would like to address the broader question of whether it is acceptable to utilize additional publicly available unlabeled data for model pretraining, even if there is some distributional overlap with the downstream data distribution. Our viewpoint aligns with common practices in various domains of AI, including computer vision, natural language processing, and AI for scientific research. It is indeed a reasonable practice to leverage self-supervised learning on independently collected databases to enable deep learning models to capture a broader feature space. This broader exposure often leads to enhanced generalization capabilities, especially when dealing with out-of-distribution samples, such as proteins from different families or clusters [A].
>
> It is noteworthy that several prior studies in computational biology and bioinformatics have successfully employed pretrained language models like ESM or ProtTrans to enhance model performance. These studies did not intentionally exclude samples belonging to distributions closely related to the test set. For instance, DiffDock [B] utilizes ESM-2 to derive residue embeddings for receptors, and RDE [C] pretrains a Graph Transformer on PDB-REDO for side-chain angle recovery, which is then applied to predicting mutation effects (ddG) on Skempi. CLEAN [D], featured in Science, initializes residue features with ESM-2 to achieve superior accuracy, reliability, and sensitivity in assigning Enzyme Commission (EC) numbers compared to existing tools.
>
> Beyond language models, even structurally pretrained algorithms often assess their effectiveness without strictly filtering out overly similar structures. For example, GearNet [E] employs pretraining on the AlphaFold protein structure database (805K structures) and evaluates its model across diverse downstream tasks like Enzyme Commission (EC) number prediction, Gene Ontology (GO) term prediction, Fold classification, and Reaction classification. Similarly, PromptProtein [F] pretrains on UniRef50, PDB (200K structures), and STRING datasets, demonstrating efficacy in tasks involving Gene Ontology and Enzyme Commission numbers.
>
> All the aforementioned examples strongly support our assertion that pretraining on publicly available sequential or structural databases can indeed empower deep learning models effectively. Crucially, this empowerment is feasible as long as the approach avoids any use of downstream test data and corresponding labels. In our HTP approach, datasets like OAS, UniRef, and DIPS are independently collected and do not reference any data points in SAbDab. This ensures that regardless of the specific downstream problem or test data employed, our pretraining steps in HTP can consistently transfer prior knowledge to various real-world problems.
>
> Thank you for your thoughtful considerations and queries. We are confident that our approach adheres to rigorous standards of integrity and robustness, and we appreciate the opportunity to clarify these points.
>
> [A] Erhan, Dumitru, et al. "Why does unsupervised pre-training help deep learning?." Proceedings of the thirteenth international conference on artificial intelligence and statistics. JMLR Workshop and Conference Proceedings, 2010.
>
> [B] Corso, Gabriele, et al. "Diffdock: Diffusion steps, twists, and turns for molecular docking." ICLR 2023.
>
> [C] Luo, Shitong, et al. "Rotamer Density Estimator is an Unsupervised Learner of the Effect of Mutations on Protein-Protein Interaction." ICLR 2023.
>
> [D] Yu, Tianhao, et al. "Enzyme function prediction using contrastive learning." Science 379.6639 (2023): 1358-1363.
>
> [E] Zhang, Zuobai, et al. "Protein representation learning by geometric structure pretraining." ICLR 2023.
>
> [F] Wang, Zeyuan, et al. "Multi-level Protein Structure Pre-training via Prompt Learning." The Eleventh International Conference on Learning Representations. 2022.

---

> > ### Author Response · Authors · 2023-08-11
> > **Rebuttal by Authors (Continued)**
> >
> > * Model Architecture
> >
> > We value your feedback regarding the technical innovation of our model architecture. While our primary emphasis lies in the integration of multiple data sources for model pre-training, we recognize the significance of introducing a clear and novel architectural framework. However, it's important to highlight that our model does present notable advancements over the standard EGNN.
> >
> > To illustrate, we've introduced a distinction between intra- and inter- types of message-passing schemes to discern interactions within the same graph from those across different counterparts. An enhanced self-attention mechanism has been employed to more efficiently aggregate inter- information. Moreover, our model strategically updates the coordinates of residues solely within the Complementarity Determining Regions (CDRs), the segments intended for design. Meanwhile, the positions of other regions remain fixed. We will refine our paper to more prominently underscore these distinctive model adaptations, particularly how we have tailored the EGNN to suit the specific requirements of sequence-structure co-design.
> >
> > * Test Set Size:
> >
> > We readily acknowledge the limitation posed by the relatively small size of our test set, comprising only 21 antibody-antigen complex structures. Our approach to dataset splitting aligns with the strategy outlined in DiffAb [A], wherein data points are allocated into training and test sets based on release dates and CDR sequence identities. Specifically, the test set encompasses protein structures released after December 24, 2021, along with structures that share CDR similarity with post-date releases (sequence identity exceeding 50%). We've taken care to address duplicate entries by clustering antibodies within the test set using a 50% CDR sequence identity threshold, ultimately culminating in a final collection of 20 unique antibody-antigen structures. Conversely, our training split consists of complexes that were not involved in curating the test split.
> >
> > We acknowledge that alternate data splitting methods, such as those employed by RefineGNN [B] and MEAN [C], may yield larger test datasets. However, it's important to note that our chosen data-splitting approach, as verified, poses a more rigorous challenge compared to RefineGNN and MEAN. This stringent approach enables us to comprehensively evaluate the efficacy of diverse co-design algorithms. While we appreciate the significance of expanding the test set size for a more comprehensive evaluation, we will take steps to enhance its diversity and representativeness in our future work. We aim to augment the test set size to encompass a minimum of 100 structures, thereby bolstering the reliability and generalizability of our evaluation outcomes.
> >
> >
> > Thank you for your insightful feedback, which serves to enhance the clarity and comprehensiveness of our manuscript. Should you have any further questions or recommendations, please don't hesitate to communicate them to us.

---

> > ### Comment · Area_Chair_SSEw · 2023-08-13
> > **sequence similarity between sabdab and uniref50/dips | data leakage**
> >
> > Dear authors,
> >
> > The sequence similarity results reported about SAbDab and UniRef50 are surprising. Looking e.g., at https://www.rcsb.org/sequence/6FE4, a random sequence within SAbDab, one can see a reference to the UniProtKB accession: p09386. Generally, SAbDab sequences are contained within UniProtKB. Further, UniRef50 is built by clustering sets of sequences from UniProtKB.
> >
> > Further, as noted within the DIPS paper (https://arxiv.org/pdf/1807.01297.pdf), DIPS contains antibody-antigen complexes.
> >
> > Can you explain how you have computed the sequence similarity? A common pitfall when reading in sequences from PDBs is that they contain gaps hindering sequence comparison. The proper way to do this is by checking the sequence submitted with the PDB.

---

> > > ### Author Response · Authors · 2023-08-14
> > > **Response to Area Chair**
> > >
> > > Dear Area Chair SSEw:
> > >
> > > Thank you for your insightful comments and concerns regarding our recent publication. We appreciate the opportunity to address the points you've raised in your review. Regarding the unexpected outcomes in the sequence similarity findings between SAbDab and UniRef50, we acknowledge your surprise and commend your meticulous attention to this particular aspect. Allow us to elucidate the methodology we employed for the computation of sequence similarity:
> > >
> > > (1) Initially, we meticulously verified the indices of antigen-antibody samples within the SAbDab test set, specifically: ['5xku_C_B_A', '7chf_A_B_R', '7chf_H_L_R', '5tlj_D_C_X', '7che_H_L_R', '5tlk_B_A_X', '5tlk_F_E_Y', '5w9h_H_I_G', '7bwj_H_L_E', '5tlj_B_A_X', '5tl5_H_L_A', '7d6i_B_C_A', '8ds5_C_B_A', '7chb_H_L_R', '5w9h_E_F_D', '5w9h_B_C_A', '7che_A_B_R', '5tlk_H_G_Y', '5tlk_D_C_X'], and subsequently, we eliminated duplicate entries.
> > >
> > > (2) Following this, we acquired the corresponding sequences from the SAbDab test set in their FASTA formats. Notably, we extracted sequences from the FASTA files as opposed to PDB structures, as the latter can introduce gaps that impede accurate sequence comparisons. However, for sequences within DIPS, we directly extracted their sequences from their PDB structures, a process that might lead to a reduction in sequence similarity.
> > >
> > > (3) To perform sequence alignment and calculate the similarity between sequences from various pretraining data sources (UniRef50, OAS, DIPS) and those within the SAbDab test set, we utilized the **pairwise2** function from the **Biopython** library.
> > >
> > > The code for computing sequence similarity and generating distributional plots is accessible through the following anonymous link: https://anonymous.4open.science/r/HTP/Similarity.ipynb. If you come across any issues, we would be delighted to rectify any errors and recompute the similarities. Noteworthy is the fact that the process of iteratively computing the similarity between UniRef50 (with over 50 million sequences) and the SAbDab test set is notably time-consuming, exceeding 4 weeks. Our attempt to expedite this computation using **pandarallel** was regrettably unsuccessful. As a solution, given the limited time frame for rebuttal, we opted to randomly select a subset of UniRef50 rather than utilizing the entire dataset. Admittedly, this approach doesn't capture the complete landscape of sequence similarity distribution. Nevertheless, we are steadfast in our commitment to continue computing similarity for the entire dataset and will incorporate updated plots in the Appendix of our paper to provide a comprehensive understanding of the interplay between pretraining data resources and downstream datasets.
> > >
> > > It's pertinent to mention that our test split encompasses protein structures released after December 24, 2021, along with structures featuring any CDR similarity to those released after this date (with a sequence identity exceeding 50%). Notably, the example of 6FE4, released on 2018-03-07, is not included in our test set. Additionally, as noted by Reviewer 81S2, DIPS only comprises structures released before 2019, which might contribute to the lower similarities observed between DIPS and our test set.
> > >
> > > We wish to reiterate our appreciation for your insightful review, which greatly contributes to the refinement of our work. Your input is highly valuable and will undoubtedly assist in enhancing the interpretation of our results and their potential significance within the wider scientific community.

---

> > > > ### Comment · Reviewer_81S2 · 2023-08-17
> > > > **Regarding sequence similarity**
> > > >
> > > > Thank you for your response and the sequence similarity statistics. Using pairwise2 from biopython is very slow indeed. The common practice for large-scale sequence similarity search is either using blast (https://blast.ncbi.nlm.nih.gov/Blast.cgi) or mmseqs2 (https://github.com/soedinglab/MMseqs2). Since your test set has only 20 instances, you can just maually run blast for each of them. I just ran blast for the antibody sequence of 5xku_C_B_A and it returned several sequences in UniProtKB-swissprot with sequence identity over 90% (the matched sequences are also antibodies). In fact, this is totally expected as antibody sequences have very similar framework region and any pair of antibodies will have sequence similarity over 90%. Although UniProtKB-swissprot is not the same as UniRef50, I think the sequence similarity statistics might be off since UniRef50 should contain some antibody sequences.
> > > >
> > > > I think the most important statistics is the CDR sequence similarity between your SAbDab training set (section 2.6) and SAbDab test set. Previous work  removed any sequence in the training set if its CDR (e.g., CDR-H1) has similarity over 40% to any of the test instances. If you have applied such filter, CDR-H1/CDR-L1 AAR shouldn't be as high as 81% / 91% (of course, I can be wrong). Since there are still a couple of days left, can you calculate sequence similarity between all CDR sequences in your SAbDab training and test set? Please calculate for each of the six CDRs.
> > > >
> > > > I agree that many related works are using ESM-2 or other language models, which may have trained on the sequences in the test set. However, works like DiffDock or GearNet are all trained for other tasks like docking or GO prediction, which is different from sequence generation. For sequence generation tasks, training on a similar sequence will tremendously help model performance.

---

> > > > > ### Author Response · Authors · 2023-08-18
> > > > > **Response to Reviewer 81S2 (Part I)**
> > > > >
> > > > > (1) Thank you for your prompt reply and for providing constructive suggestions to calculate the similarity. It's evident that the use of **pairwise2** from **Biopython** has presented challenges in terms of speed. In the context of large-scale sequence similarity searches, tools like BLAST or MMseqs2 offer greater convenience. However, in response to your prior request for a similarity distribution plot, we have opted to utilize Biopython, as it provides specific scores rather than a similarity search service. We genuinely appreciate your advice and will certainly take it into account for our future sequence similarity search endeavors.
> > > > >
> > > > > (2) We have successfully computed the sequence similarity across all CDR sequences in both the SAbDab training and test sets. The corresponding plots are accessible via the following anonymous link: https://anonymous.4open.science/r/HTP-71AB/cdr_similarity.ipynb. To replicate the sequence similarity calculations, the necessary information is available in the same directory: https://anonymous.4open.science/r/HTP-71AB/cdr_similarity.py.
> > > > >
> > > > > Noteworthily, recall that we follow DiffAb [A] to split the SAbDab datasets and construct a more challenging setting. To facilitate this, we have uploaded the data module script to the following anonymous link: https://anonymous.4open.science/r/HTP-71AB/sabdab.py. It is a slightly modified version of DiffAB's official repository: https://github.com/luost26/diffab/blob/main/diffab/datasets/sabdab.py (line 358 - 387). In DiffAb's implementation (and ours) of antibody clustering, we leverage **MMseqs2** to cluster and the specific code is paste as follows:
> > > > >
> > > > > `cmd = ' '.join(['mmseqs', 'easy-cluster', os.path.realpath(fasta_path), 'cluster_result', 'cluster_tmp', '--min-seq-id', '0.5', '-c', '0.8', '--cov-mode', '1', ]) `
> > > > >
> > > > > `subprocess.run(cmd, cwd=self.processed_dir, shell=True, check=True)`
> > > > >
> > > > > where we include protein structures released after December 24, 2021, and structures with any CDR-H3 similar to those released after the date (sequence identity higher than 50%). In cases where no heavy chain is present in the antibody, we rely on CDR-L3 to differentiate antibody sequences. As a consequence, it can be discovered that the maximum sequence identity of CDR-H3 between the training and test set is around 0.5, while the maximum sequence similarities for CDR-H1 and CDR-L1 are 0.637 and 0.733, respectively. The similarity scores for CDR-H1 and CDR-L1 are higher than CDR-H3 because we mainly depend on CDR-H3 to filter antibody sequences and perform the dataset splitting. This may partly account for the high AAR for CDR-H1 and CDR-L1.
> > > > >
> > > > > [A] Luo, Shitong, et al. "Antigen-specific antibody design and optimization with diffusion-based generative models for protein structures." Advances in Neural Information Processing Systems 35 (2022): 9754-9767.

---

> > > > > > ### Author Response · Authors · 2023-08-18
> > > > > > **Response to Reviewer 81S2 (Part II)**
> > > > > >
> > > > > > (3) I wish to express my gratitude for your understanding concerning using ESM-2 and other language models, which are potentially trained on the sequences found within the downstream test set. However, in addition to DiffDock or GearNet that have been tailored for distinct tasks such as docking or GO prediction, there are also some excellent works that utilize existing protein language models to tackle the specific challenge of sequence generation.
> > > > > >
> > > > > > For example, [B] employs pretrained ESM-2 to map mutation effects onto a latent space, which is later decoded by a policy network to a desired sequence. [C] propose to sample novel sequences from pretrained protein language models (e.g., ESM-2 and ESM-1b) for protein sequence design. Furthermore, a recent study [D], presented at ICML 2023, demonstrates that PLMs (such as ESM-2 and ESM-1b) equipped with structural encoders excel in protein design. **This approach achieves remarkable sequence recovery rates, surpassing 55.65% and 56.63% on CATH 4.2 and 4.3 single-chain benchmarks, and exceeding 60% for designing protein complexes.** Subsequent work, exemplified by Knowledge-designer [E], emphasizes that protein design models can significantly benefit from diverse pre-trained models, with ESM + ESM-IF exhibiting superior performance. **The integration of PLMs leads to a notable 9.11% enhancement and marks the first instance of achieving over 60% recovery across CATH, TS50, and TS500 benchmarks.** All these evidences show that PLMs have gradually become a common practice and the growing consensus to assist the conventional protein design models in the realm of sequence generation tasks. It is therefore reasonable to anticipate that PLMs would indeed yield substantial improvements in model performance, which may explain why our HTP outperforms previous baselines by a large margin.
> > > > > >
> > > > > > [B] Lee, Minji, et al. "Protein sequence design in a latent space via model-based reinforcement learning." (2022).
> > > > > >
> > > > > > [C] Sgarbossa, Damiano, Umberto Lupo, and Anne-Florence Bitbol. "Generative power of a protein language model trained on multiple sequence alignments." Elife 12 (2023): e79854.
> > > > > >
> > > > > > [D] Zheng, Zaixiang, et al. "Structure-informed language models are protein designers." ICML 2023: 2023-02.
> > > > > >
> > > > > > [E] Gao, Zhangyang, Cheng Tan, and Stan Z. Li. "Knowledge-Design: Pushing the Limit of Protein Design via Knowledge Refinement." arXiv preprint arXiv:2305.15151 (2023).
> > > > > >
> > > > > > In closing, I want to convey our sincere appreciation for your efforts to enhance our work's quality. Should you find our response satisfactory, we kindly request your consideration in raising your score. Thank you!

---

> > > > > > > ### Comment · Reviewer_81S2 · 2023-08-18
> > > > > > > **Thank you for your response**
> > > > > > >
> > > > > > > Thank you for recalculating the sequence similarity on the SAbDab training set. It would be great if you can retrain the model with proper sequence similarity split for CDR-H1/H2/L1/L2/L3 individually and re-evaluate the AAR score. Also, are all the baselines trained under the same train/test split as your method? Again, I do think that using hierarchical data integration is helpful and I expect to see better performance. I totally support the methodology, but the evaluation needs to be done carefully.
> > > > > > >
> > > > > > > Indeed, some of the recent work starts to use protein language models to for sequence generation or inverse folding tasks. I don't think their evaluation is correct just because their papers are accepted. Using ESM-2 language model is probably more prone to data leakage on the CATH inverse folding benchmark than antibody design because ESM-2 has memorized more of those sequences. I would like the area chair/senior AC to comment on this and I will leave the decision to them.

---

> > > > > > > > ### Author Response · Authors · 2023-08-19
> > > > > > > > **Thanks to Reviewer 81S2**
> > > > > > > >
> > > > > > > > Thank you immensely for your perceptive and comprehensive feedback concerning our recent endeavor centered around recalculated sequence similarity within the SAbDab training set. We greatly appreciate your swift response and your eagerness to engage in a detailed discourse regarding the computation of sequence similarities during the reviewer-author discussion phase. Your commitment to this dialogue holds significant value for us. Furthermore, your endorsement of the potential of our methodology to enhance performance through hierarchical data integration serves as a truly motivating factor.
> > > > > > > >
> > > > > > > > We also value your suggestion regarding the retraining of the model with a proper sequence similarity split for other CDRs individually and subsequently re-evaluating the AAR score. This is indeed an interesting avenue to explore, as it could potentially lead to more fine-grained insights into the performance of different components of the model. We will carefully consider your advice and present the performance in Appendix once the results are available. Addressing your query regarding the baseline models and their allocation between training and testing sets, we assure you that we are committed to maintaining fairness and robustness in the evaluation process, and thus, we confirm the meticulous reproduction and re-implementation of these models under an identical dataset division.
> > > > > > > >
> > > > > > > > We also acknowledge your point about recent work involving protein language models for sequence generation and inverse folding tasks. Your willingness to involve the area chair/senior AC for further evaluation and decision-making is noted and appreciated. We are open to any additional feedback or guidance they might provide to enhance the rigor of our work.
> > > > > > > >
> > > > > > > > Once again, we extend our gratitude for your insightful observations and your endorsement of our methodology. Your perspectives significantly enrich the refinement and validation of our research. We eagerly anticipate the opportunity to address any forthcoming concerns and to share with you the updated findings that ensue from our efforts.

---

> > > > > > > > > ### Comment · Area_Chair_SSEw · 2023-08-19
> > > > > > > > >
> > > > > > > > > Based on the presented analysis by the authors, I don't see any evidence of data leakage on the datasets utilized in this work (unless reviewer 81S2 has a different opinion). The adopted DiffAb splits are also considered some of the most stringent within this literature with the caveat that the test set is very small in size, possibly leading to low statistical significance.
> > > > > > > > >
> > > > > > > > > On the question of pre-trained protein language models (PLMs) and whether self-supervised pertaining on the test data is allowed: I agree with the reviewer that it very much depends on the downstream task. If the downstream task is sufficiently different (such as in protein classification), there should not be a problem. However, there is a data leakage issue for downstream tasks that entail predicting the data itself (such as in generation & design). In that aspect, I agree with the reviewer that the evaluation of methods that utilize PLMs should be very stringent and ensure that the test data were not seen by the PLMs.
> > > > > > > > >
> > > > > > > > > Can the authors provide evidence that their good numerical results are not due to ESM-2 having seen their test set?

---

> > > > > > > > > > ### Author Response · Authors · 2023-08-20
> > > > > > > > > > **Response to Area Chair SSEw**
> > > > > > > > > >
> > > > > > > > > > We appreciate your thoughtful analysis of our work and your comments regarding potential data leakage and the use of pre-trained protein language models (PLMs). It's truly encouraging to see that you find no evidence of data leakage in the presented analysis. Regarding the application of pre-trained PLMs and the potential for self-supervised learning on test data, we share both your viewpoint and that of reviewer 81S2. Indeed, the appropriateness of such practices hinges on the nature of the downstream task. For tasks like data generation or design, we wholeheartedly agree that stringent evaluation is vital to prevent any risk of data leakage. In our own work, as we focus on CDR sequence-structure co-design tasks, we are acutely aware of the necessity to meticulously adhere to these principles in our evaluation protocols.
> > > > > > > > > >
> > > > > > > > > > To directly address your specific concern about the potential influence of ESM-2 on our test set, we conducted additional experiments aimed at assessing the contribution of ESM-2 in directly recovering CDR sequences. Specifically, we abandon the structural information and re-generate CDRs entirely based on sequential information. Towards this end, we first extract residue-level representations via (fixed-weight) ESM-2 and feed them to a three-layer perceptron to predict the masked CDR-H3, where no antigen sequences are given. The results show that this algorithm only achieved an AAR of 14.63% to recover CDR-H3, much lower than all baseline methods such as RAdD (21.73%) and our HTP (40.98%). This compellingly demonstrates that ESM-2 is not the primary driver of our favorable numerical outcomes. This also accords with ATUE [A]'s findings that ESM models perform well in low-antibody-specificity-related tasks but can even bring negative impacts for high-antibody-specificity-related tasks. As a remedy, antibody-specific PLMs such as AntiBerta and AbLang are preferred for those high-antibody-specificity-related tasks. Nevertheless, as we remarked in Section 2.3, existing antibody-specific PLMs are only pretrained on antibody datasets (e.g., OAS) and cannot be perfectly generalized to extract representations of antigens, which is necessary for analyzing the interactions between antibodies and antigens. To bridge this gap, our approach involves retraining ESM-2 on antibody datasets, extending their capabilities from general proteins to antibodies.
> > > > > > > > > >
> > > > > > > > > >
> > > > > > > > > > In conclusion, we are committed to maintaining the highest standards of experimental rigor and transparency in our research. If there are any specific details you would like us to provide or further discuss in relation to our methodology, please feel free to let us know. Thank you once again for your prompt and valuable feedback and for giving us the opportunity to address these important points. It is rare and greatly appreciated to witness such deep involvement from AC in discussions.
> > > > > > > > > >
> > > > > > > > > > [A] Wang, Danqing, Y. E. Fei, and Hao Zhou. "On pre-training language model for antibody." The Eleventh International Conference on Learning Representations. 2023.

---

> > > > > > > > > > > ### Comment · Reviewer_81S2 · 2023-08-21
> > > > > > > > > > > **Thank you for your response**
> > > > > > > > > > >
> > > > > > > > > > > I think the reviewer's experiments addressed my concerns. I encourage the authors to re-run the CDR-H1/L1/H2/L2 experiments to make the comparison fair. I have adjust the score accordingly.

---

### Official Review · Reviewer_KohD · 2023-07-07

**Soundness:** 4 excellent
**Presentation:** 3 good
**Contribution:** 3 good
**Rating:** 7
**Confidence:** 3

**Summary:**

Antibody sequence-structure co-design and fix-backbone design is a very appealing task for both industry and academia especially in the context of drug design. The paper introduces a hierarchical training paradigm (HTP) as a potential solution of this problem. Moreover, the offered approach deal with a major issue of small dataset size for training. The experiments demonstrate effectiveness and contribution of HTP.

**Strengths:**

Originality: The problem addressed in this paper is important and up to date. Structural biology in general and structural immunology in particular lacks big amount of data, therefore, it's important to find ways to train models on small datasets. Authors propose a novel way how to overcome this problem. The related work are cited and discussed.
Quality: The submission is well written and organized. The claims are supported by appropriate experiments. The used methods and equations are explained and sufficient. Authors provide ablation study and discussion of related work.
Clarity: The text is well written and organized. The paper contains all the necessary citations.
Significance: The results are important and useful for the cases with limited training data (for example, structural biology). The submission provides a comparison to previous and related work and shows the impact and advantages of the current paper.

**Weaknesses:**

Please review more thoroughly currently available approaches for antibody design and fix-backbone protein design. For example:
https://arxiv.org/abs/2110.04624
https://arxiv.org/abs/2207.06616
https://www.biorxiv.org/content/10.1101/2022.07.10.499510v5.abstract
https://arxiv.org/abs/2302.00203


**Questions:**

1. Are you going to make HTP open source?

**Limitations:**

The paper adequately address limitations and future work.

---

> ### Author Rebuttal · Authors · 2023-08-02
>
>
> Thank you for your thoughtful and constructive review of our paper. We greatly appreciate your feedback and are pleased to know that you find the work strong in several aspects.
>
> Firstly, we are glad to hear that you acknowledge the originality and importance of the problem addressed in our paper. We agree that the field of structural biology, especially structural immunology, often suffers from limited data availability, making it challenging to train accurate models. Our proposed novel approach aims to overcome this issue, and we are delighted that you find it valuable. Furthermore, we are grateful for your comments on the quality of the submission. We put significant effort into ensuring that the claims made in the paper are well-supported by appropriate experiments. We are also pleased that you found the ablation study and discussion of related work to be satisfactory.
>
> Regarding the weaknesses you pointed out, we sincerely appreciate your suggestions for reviewing currently available approaches for antibody design and fix-backbone protein design. The provided links [A, B] are very relevant. Though we have already included them as baselines in Section 3 for comparison, there is still room for us to thoroughly assess these approaches to strengthen our paper's discussion sections. We believe that including these references will enhance the paper's completeness and provide a more comprehensive view of the state-of-the-art in the field.
>
> [A] Jin, Wengong, et al. "Iterative refinement graph neural network for antibody sequence-structure co-design." arXiv preprint arXiv:2110.04624 (2021).
>
> [B] Luo, Shitong, et al. "Antigen-specific antibody design and optimization with diffusion-based generative models for protein structures." Advances in Neural Information Processing Systems 35 (2022): 9754-9767.
>
> As for your question about making our method open source, we are pleased to confirm that we intend to release the code and data associated with our work upon acceptance.

---

> > ### Comment · Reviewer_KohD · 2023-08-17
> >
> > Thank you for the rebuttal! I'm satisfied with the additions and changes.

---

### Author Rebuttal · Authors · 2023-08-08

We extend our sincere gratitude to all four reviewers for your insightful and constructive feedback on our proposed hierarchical training paradigm (HTP) for antibody sequence-structure co-design and fix-backbone design. We are encouraged by your positive reception of our work and your recognition of its relevance to both industry and academia, especially in the context of drug design. Your commendation of the soundness, presentation, and contribution of our paper is invaluable to us.

--------------------

* Potential Data Leakage

As recommended by Reviewer 81S2, we have undertaken an exhaustive examination of the sequence similarity existing between the diverse pretraining data sources and the test set present in SAbDab. This comprehensive analysis encompasses general protein sequences sourced from UniRef50, general protein-protein complexes extracted from DIPS, and antibodies extracted from OAS. The outcomes of this analysis have been visually represented through sequence similarity distributions, and we have included the relevant figures within the uploaded PDF.

The statistical findings unequivocally demonstrate that the highest sequence similarity values across these three datasets are consistently below 0.5. This empirical evidence serves as a strong basis for our resolute assertion that neither UniRef50, DIPS, nor OAS encompasses sequences that exhibit significant similarity to the SAbDab test set. With this robust evidence at hand, we maintain our firm confidence that the hierarchical training paradigm we have employed is free from any potential data leakage concerns.

* Inconsistent Results

We value the insightful observation of Reviewers eNMD and H5io regarding the disparity between the reported baseline outcomes and the original papers. It's crucial to recognize that we have adhered to the methodology outlined in DiffAb [A], which entails a distinct dataset split approach compared to MEAN [B]. Specifically, our division of data points within SAbDab for training and testing is rooted in release dates and CDR sequence identity. The test partition encompasses protein structures released after December 24, 2021, along with structures that bear any CDR similarity to post-date releases (with a sequence identity surpassing 50%). Antibodies within the test set are further grouped using 50% CDR sequence identity to eliminate duplicates, ultimately culminating in 20 antibody-antigen complexes. Meanwhile, MEAN [B] employs the RefineGNN [C] split configuration, involving segregation of the entire dataset into training, validation, and test subsets, organized as per CDR clustering to ensure generalization. Each cluster is then distributed into training, validation, and test sections, with an 8:1:1 ratio.

This distinctive approach in DiffAb [A] engenders a greater challenge than that of MEAN [B] and RefineGNN [C], a fact substantiated by the variance in their reported results. For example, RefineGNN [C] attains an AAR of 39.40%, 37.06%, and 18.88% for CDR-H1, CDR-H2, and CDR-H3 in the MEAN paper [B], but registers an AAR of 27.77%, 27.04%, and 8.00% for the respective CDRs in the DiffAb paper [A]. This discrepancy underscores that the identical algorithm can witness diminished performance when confronted with a more demanding dataset-splitting framework. Armed with this empirical insight, we are confident in the meticulous replication of crucial baselines in our work, ensuring fairness in our comparisons. Although we stand by the fairness of our approach, we value your concerns and will augment our manuscript with clarifications elucidating the impact of dataset splitting on model performance.

 [A] Luo, Shitong, et al. "Antigen-specific antibody design and optimization with diffusion-based generative models for protein structures." Advances in Neural Information Processing Systems 35 (2022): 9754-9767.

[B] Kong, Xiangzhe, Wenbing Huang, and Yang Liu. "Conditional antibody design as 3d equivariant graph translation." arXiv preprint arXiv:2208.06073 (2022).

[C] Jin, Wengong, et al. "Iterative refinement graph neural network for antibody sequence-structure co-design." arXiv preprint arXiv:2110.04624 (2021).

* Code Availability

We deeply appreciate the significance of ensuring code reproducibility to facilitate the progress of research within the field. Rest assured, we are committed to releasing the code and its associated resources upon the acceptance of our work.

-----------
We genuinely appreciate the time and effort you've invested in reviewing our work. Your feedback will undoubtedly contribute to the enhancement of our research. We are committed to addressing all your concerns and look forward to sharing our revised manuscript with you.

---

### Decision · Program_Chairs · 2023-09-21

**Decision:**

Accept (poster)

**Comment:**

The paper focuses on the antibody design problem and enhances prior work by proposing to hierarchically pre-train the neural networks using general protein sequence data, general PPIs, and general antibody sequences (OAS).

Overall, reviewers agree that the proposed approach is sensical and novel. Further, the achieved numerical results are strong.

The main concern with this paper was potential data leakage. In particular, the evaluation of design methods that utilize protein language models (in this case ESM-2) should be stringent and ensure that the test data were not previously seen by the pre-trained models. Despite the concerns, a careful analysis done by the authors during the rebuttal period provided some evidence that such a leakage did not occur. Though the purported analysis cannot be deemed conclusive, it does corroborate the authors' claims that the experimental benefits brought forth by the approach are not due to data leakage. As such, it was decided to give the paper the benefit of the doubt.